

The Probabilistic Drought Forecast Based on the Ensemble Technique Using
the Korean Surface Water Supply Index
Suk Hwan Jang[1], Jae-Kyoung Lee[2], Ji Hwan Oh[3], Jun Won Jo[4], and Younghyun Cho[5]
1. Professor, Department of Civil Engineering, Daejin University, Pocheon-si, Gyeonggi-do,
Korea, drjang@daejin.ac.kr
2. Assistant Professor, Innovation Center for Engineering Education, Daejin University,
Pocheon-si, Gyeonggi-do, Korea, myroom1@daejin.ac.kr
3. Ph.D Candidate, Department of Civil Engineering, Daejin University, Pocheon-si,
Gyeonggi-do, Korea, ojh4525@naver.com
4. Master course, Department of Civil Engineering, Daejin University, Pocheon-si,
Gyeonggi-do, Korea, yhjowon@naver.com
5. Principal Researcher, Hydrometeorological Cooperation Center, Gwacheon-si, Gyeonggi-
do, Korea, yhcho@kwater.or.kr

22  Corresponding Author: Jae-Kyoung Lee, Innovation Center for Engineering Education,

23  Daejin University, Pocheon-si, Gyeonggi-do, Korea

24  E-mail : myroom1@daejin.ac.kr



1                              Abstract

This study proposes the new hydrological drought index, Korean Surface Water Supply Index
(KSWSI), which overcomes some of the limitations in the calculation of previous SWSI applied in
Korea and conducts the probabilistic drought forecasts using KSWSI. In this study, all
hydrometeorological variables in the Geum River basin were investigated and appropriate variables
were selected as KSWSI components for each sub-basin. And whereby only the normal distributions
are applied to all drought components, probability distributions suitable for each KSWSI component
were estimated. As a result of verifying KSWSIs, the accuracy of KSWSIs showed better drought
phenomenon in drought events. The monthly probabilistic drought forecasts were also calculated
based on ensemble technique using KSWSI. In 2006 and 2014 drought events, the accuracy of the
drought forecasts using KSWSIs were higher than those using previous SWSI, demonstrating that
KSWSI is able to enhance the accuracy of drought forecasts. The influence of expanding
hydrometeorological components and selecting appropriate probability distributions for each KSWSI
component were analyzed. It is confirmed that the accuracy of KSWSIs may be affected by the choice
of hydrometerological components, the station data obtained, the length of used data for each station,
and the probability distributions selected. Furthermore, the uncertainty quantification of the KSWSI
calculation procedure was also carried out using the Maximum Entropy (ME) theory. The large MEs
and standard deviations of KSWSIs in the flood season cause uncertainties, implying that the selection
of the appropriate probability distributions for selected drought components in the flood season is
very important.
Key words: Hydrological drought, Korean surface water supply index, Probabilistic drought forecast,
Uncertainty quantification, Maximum entropy





# 1. INTRODUCTION

From 2014 to 2015, a great deal of economic damage has occurred because of the shortage of agricultural water due to drought throughout the Korean Peninsula, especially in the northern part of Gyeonggi-do. Droughts have dramatic impacts on the socio-economic state and their occurrence is becoming more frequent. Drought management is difficult not only because of the seasonal characteristics (which means that more than 60% of the annual average rainfalls occur in the summer season), but also because of the dry flood season caused by global warming. The water shortage stresses the small agricultural and municipal water reservoirs, making it difficult to manage water resources plans and policies (Choi, 2002). In order to effectively mitigate these droughts, continual improvement of drought indexes should be prioritized; because the drought occurs due to various conditions and circumstances, it is difficult to reflect all of these in the drought index. The various drought indices used in Korea have some problems as follows: determining the hydrological meteorological factors to be utilized, determining whether the improved or developed drought index can be extended in all regions, and determining how to set thresholds to distinguish among the stages of the drought index. These considerations make it difficult to accurately monitor and forecast actual droughts.

In hydrological drought, the effects of hydrological variables on drought such as streamflow, soil water, and groundwater are physically delayed compared to meteorological variables such as precipitation and evapotranspiration, so that these characteristics can be reflected in the hydrological drought index. Recently, various hydrological drought indices have been developed and improved. Shukla and Wood (2008) developed the Standardized Runoff Index (SRI) using hydrological variables and contrast results of a SRI with that of a Standardized Precipitation Index (SPI) during drought events in a snowmelt region. Karamouz et al. (2009) developed an integrated index, the Hybrid Drought Index (HDI), which was combined with the well-known SPI, Water Surface Supply Index (SWSI), and Palmer Drought Severity Index (PDSI) and applied to the Gavkhooni/Zayandeh-rud basin in the central part of Iran. Karamouz et al. concluded that the results of the HDI show its significant value for drought prediction. Dogan et al. (2012) compared and analyzed six different





drought indices for intensity and duration of drought in Kenya, and concluded that the Effective
Drought Index (EDI) was consistent with other drought indices for various time-steps and was
preferable for monitoring long-term droughts in arid/semi-arid regions. Ahn and Kim (2010)
developed the Water Ability Index (WAI) based on the amount of water available in a basin, which
could replace the SWSI as a hydrological drought index in Korea. Park et al. (2011) then proposed the
Water Availability Drought Index (WADI) to improve the shortcomings of previous domestic
hydrological indices which did not reflect water supply and water intake or reservoir and dam
facilities.
Drought forecast should also be performed in preparing for drought and creating proactive
drought policies and preparedness plans. White et al. (2004) utilized the optimized Canonical
Correlation Analysis (CCA) to forecast principal components of summer precipitation anomalies to
predict the duration of drought over eastern and central Australia. Belayneh and Admowski (2013)
proposed the use of three machine learning techniques, Artificial Neural Network (ANN), Support
Vector Regression (SVR), and coupled WAvelet-ANNs (WA-ANN), to forecast short-term drought
for short lead times with SPI in the Awaash river basin of Ethiopia. The results revealed that the WA-
ANN model was the most accurate for forecasting SPI3 and SPI6 values over lead times of one and
three months. Son and Bae (2015) reviewed the availability of the Ensemble Streamflow Prediction
(ESP) technique for hydrological drought forecasting and showed that it is effective for a 1-2 months
outlook in Korea. However, studies of domestic drought forecast are only at the beginning stage, and
projected meteorological data is necessary for drought forecast. However, it is difficult to utilize the
data due to the uncertainty of the future projected meteorological data and the limitation of data
acquisition and connection.
Therefore, this study proposes the new and improved hydrological drought index and conducts
the methodology to forecast droughts in future for the Korean Peninsula as follows: Firstly, in this
study, the limitations of the existing hydrological drought index are analyzed, improvements are made,
and a new drought index is applied called the Korean Water Surface Supply Index (KSWSI).
Secondly, the probabilistic monthly drought forecasts are estimated using the KSWSI based on the





ensemble technique. Lastly, the effect of the selection of drought components and their probability
distributions is analyzed and a method is proposed to quantify their uncertainties.
2. Methodology
2.1 Study basin
This section describes the Geum River basin as the applicable area for improving the drought
index and verifying the drought forecast (Fig. 1). The Geum River basin flows north-westerly to about
its mid-point, then generally south-westerly for 401km. It consists of 21 sub-basins, and drains into an
area of 9,810km$^2$. The Geum River basin has two multi-purpose dams, Daecheong Dam and
Yongdam Dam. Daecheong Dam provides municipal and industrial water supply to Daejeon and
Chungju, and Yongdam Dam (which is only one-fifth the size of the Daecheong Dam drainage area)
supplies water to Jeonju. Analyzing the river flow in the Geum River basin is relatively simple
because it has fewer dams and a simpler river system than other basins. The region of the Geum River
basin has been affected by considerable drought since 2000 year and has been widely used in previous
drought studies in Korea.

17           [Fig. 1. Study basin: 14 sub-basins in Geum River basin]

2.2 Improvement of hydrological drought index: Surface Water Supply Index (SWSI)
The Surface Water Supply Index (SWSI) (Shafer and Dezman, 1982) was selected as the well-
known hydrological drought index. SWSI is advantageous as it can flexibly utilize various
hydrometeorological components depending on the study basin. SWSI is based on probability
distributions of monthly time series of individual component indices and is calculated using four
hydrometeorological components: snowpack, precipitation, streamflow, and reservoir storage. It is
also an appropriate drought indicator in snow-dominated regions. The drought classification of SWSI
is divided into seven categories (extremely dry (-4.2 to -3.0; 7$^{th}$ category), moderately dry (-2.9 to -2.0;





6th category), slightly dry (-1.9 to -1.0; 5th category), near average (-0.9 to 1.0; 4th category), slightly
wet (1.1 to 2.0; 3rd category), moderately wet (2.1 to 3.0; 2nd category), and extremely wet (3.1 to 4.2;
1st category)) and is similar to the typical categories of the Palmer Drought Severity Index (PDSI).
The mathematical formulation of SWSI is given by:

$$\text{SWSI}_t = \frac{w_1 P_t^{snow} + w_2 P_t^{prec} + w_3 P_t^{strm} + w_4 P_t^{resv} - 50}{12} \tag{1}$$

where $w_1$, $w_2$, $w_3$, and $w_4$ are the weights for each hydro-meteorological component and $w_1 + w_2 + w_3 +$
$w_4 = 1$, and where $t$ represents the monthly time-step. $P_t^i$ is the non-exceedance probability (in
percentage) for component $i$ where the superscripts of *snow*, *prec*, *strm*, and *resv* represent the
snowpack, precipitation, streamflow, and reservoir storage in time $t$, respectively. In calculating the
SWSI, depending on regions, a snowpack component is applied from December to the subsequent
May, and a streamflow component is applied during the remaining periods. Kwon et al. (2006) and
Kwon and Kim (2006) then developed a Modified SWSI (called MSWSI) by improving SWSI for the
Korean Peninsula. In MSWSI, the snowpack parameter is replaced by groundwater because the
portion of underground water is more important to snowpack in the water resources management in
Korea:

$$\text{MSWSI}_t = \frac{w_1 P_t^{gw} + w_2 P_t^{prec} + w_3 P_t^{strm} + w_4 P_t^{resv} - 50}{12} \tag{2}$$

where *gw* represents the groundwater component. The process of MSWSI calculation is as follows:
Step 1: Analysis of available hydrometeorological components by basins
Step 2: Selection of available hydrometeorological components and collection of observed data
Step 3: Calculation of weights for each hydrometeorological component


Step 4: Estimation of probability distributions for each hydrometeorological component
Step 5: Calculation of MSWSI values using Eq. (2)
However, this process of MSWSI calculation has several limitations. Firstly, only four
hydrometeorological components are used in the previous MSWSI calculation in Steps 1 & 2 and the
MSWSI is not able to reflect more various components. Different hydrometeorological components
actually impact drought events depending on data length, the urban area, and upstream & downstream
areas of dams; therefore, the available components should be widely investigated. Secondly, in Step 4,
probability distributions of all components were fitted to the only normal distribution in the MSWSI
calculation process. Estimating the appropriate probability distribution for each component yields
accurate non-exceedance probability values, which can be used to estimate the near actual drought
index.
Therefore, in this study, an improved MSWSI was developed, called the Korean SWSI (KSWSI),
with two improvements. The first improvement involves investigating all available
hydrometeorological components and selecting the appropriate components for each sub-basin. The
second improvement involves estimating and applying a suitable probability distribution for each
selected hydrometeorological component. The detailed improvements are as described in the
following section and Fig. 2 shows the process of the MSWSI calculation and its improvements.
*Investigation and selection of available hydrometeorological components*
In this section, all hydrometeorological data from each sub-basin in the Geum River basin were
investigated and classified into 9 types: precipitation data, water level data in dam, meteorological
data, national streamflow data, local streamflow levels 1 & 2 data, multi-regional water supply, local
water supply, and groundwater (Table 1(a)). The precipitation data, water level data, water discharge
data, streamflow data, dam data (included in inflow, release, and storage data), and groundwater data
were selected as practical hydrometeorological components on the basis of ease of data acquisition,
data quality control, and data length. These data were then collected from each observation station





(Table 1(b)). Table 2 shows the final hydrometeorological components selected for each sub-basin.
The values of the previous MSWSI are calculated using precipitation data obtained from six stations,
streamflow data obtained from 10 stations, groundwater data obtained from 3 stations, and dam
inflow data for calculating the values of KSWSIs in this study. However, the types of data were
extended to include (areal-averaged) precipitation data from 42 stations, streamflow data from 28
stations, groundwater data from 7 stations, and dam data included in inflow, release, and storage data.
The sub-basins were also classified into upstream of dam, downstream of dam, streamflow,
groundwater, precipitation, and water supply-dominant basin depending on the most influential
hydrometeorological component that has the largest weight for each sub-basin. Doesken et al. (1991)
proposed a method that can reflect the relative contribution of hydrometeorological components to
estimate the weights ($w_1$, $w_2$, $w_3$, and $w_4$). The initial weights of each month for each component were
calculated as monthly values divided by the annual total of the component. The calculated monthly
values of selected components of KSWSI were summed for each month. Then, the twelve monthly
sums, calculated using this procedure, were divided by their total sum to find the sum of the final
weights as 1. As shown in Fig. 3, sub-basins adjacent to the upstream and downstream of dams were
affected by components related to dam data and streamflow and precipitation components had an
important impact in other sub-basins. Especially, the effects of streamflow and precipitation
components are varied slightly month by month, with the effect of the precipitation component being
greater in the flood season.
*Improvement of probability distribution estimation for each hydrometeorological component*

22        In this section, the probability distributions (Generalized Extreme Value (GEV), Gumbel, normal,

2-parameter log-normal, log-normal, and 3-parameter log-normal distribution) applicable to each
hydrometeorological component and parameter estimation methods (e.g. maximum likelihood method,
probability weighted moment method, and method of moment) are applied and then log-likelihood
test is also used for the goodness of fit test. Table 3 shows final selected probability distributions for
each hydrometeorological component and sub-basin.





[Fig. 2. Procedure of KSWSI calculation and two improvements proposed by this study]
[Table 1. Basic investigation of hydrometerological components at each sub-basin]
[Table 2. Selected hydrometerological components and stations at each sub-basin]
[Fig. 3. Example of weights of each hydrometeorological variable for each month at sub-basin
3001 and 3007]
[Table 3. Selected suitable probability distributions to hydrometerological components at each sub-
basin]
2.3 Application results of KSWSI
In this study, 2001, 2006, and 2014-year events were used, when the severe drought occurred
nationally. In the 2001 event, the average rainfall amounts were as high as 377mm from March to
May, which was 20%~30% of the annual rainfall amounts in some regions in Korea. The rainfall
amounts from August to October was only 30% of the annual rainfall amounts in the south part of the
Korean Peninsula in 2006 and the national reservoir storage ratio was 67% on average (NEMA, 2009).
In 2014, a severe drought occurred in northern Korea, where average rainfall amounts were 50%~61%
compared to the normal-year, where the normal-year is the mean of the last 30-year average rainfall
(KMA, 2014).
Fig. 4 shows the results of the MSWSI and KSWSI for April in 2001, 2006, and 2014 in Geum
River basin. In 2001, both MSWSI and KSWSI generally showed a similar drought trend; while the
MSWSI in the Daecheong Dam had moderate and extreme droughts, the KSWSIs showed near
normal and slight droughts. In 2006 and 2014, the KSWSIs showed stronger drought intensities in
some sub-basins than the MSWSIs; especially, KSWSIs indicated that droughts in the western sub-
basins were more severe.
*Comparison of MSWSIs and KSWSIs in sub-basin 3001*





Fig. 5(a) shows the time series for the MSWSIs and the KSWSIs in sub-basin 3001 for the 2014
event. In the MSWSI, slightly severe or severe droughts were simulated to occur continuously;
however, KSWSIs were overall above the near normal droughts. Fig. 5(b) shows the time-series of
non-dimensional ratios to the normal droughts during in the 2013-2014 years for each
hydrometeorological component such as precipitation, streamflow, and dam inflow. In block A of Fig.
5(a), the ratios of precipitation and dam inflows were lower than the normal-year in January-February
2014, but inflows and streamflow were abundant due to the increased precipitation (up to 164%)
compared to the normal-year from September to December 2013. As these effects continued until
early 2014, it is more reasonable to assume that hydrological drought did not occur in sub-basin 3001.
In the flood season, the amount of precipitation and dam inflow were lower than the normal-year, but
water shortage did not occur due to the abundant precipitations from March to April. In block B of Fig.
5(a), MSWSI showed sub-basin 3001 under drought conditions, but the dam inflow and streamflow
increased due to the significantly higher precipitation than normal-year, and KSWSIs showed that
sub-basin 3001 was more moderately wet.
*Comparison of MSWSIs and KSWSIs in sub-basin 3014*
Fig. 5(c) shows the time series for the MSWSIs and the KSWSIs in sub-basin 3014 for the 2001
event. The MSWSIs were somewhat varied; however, most of them were above the normal drought
level and no dry condition occurred, except in July and August. On the other hand, in the KSWSIs,
most of the droughts occurred in 2001, and severe drought occurred in early 2001. Fig. 5(d) shows the
time-series of the non-dimensional ratios to the normal-year during the 2001-2002 years for each
hydrometeorological component such as precipitation, streamflow, and dam inflow. During the block
period shown in Fig. 5(c), the amount of precipitation and streamflow, which were only 40%~60% of
the normal-year, contributed to the water storage, resulting in severe drought (Fig. 5(d)). Therefore, it
is more reasonable to conclude that hydrological drought occurred in sub-basin 3014.





As shown in the previous examples, compared to the MSWSIs, the KSWSIs calculated more
accurate drought results in the Geum River basin. Therefore, it is confirmed that the KSWSI is more
appropriate in hydrological drought monitoring and forecasting.
[Fig. 4. Comparison of the MSWSI and KSWSI results in April 2001, 2006, and 2014 years]
[Fig. 5. Verification of KSWSI results in sub-basin 3001 and 3014]
3. Probabilistic Drought Forecasts with the KSWSI
3.1 Monthly drought forecasts based on ensemble technique
*Application outline*
This study considered 16 historical scenarios (1990~2005) and 24 historical scenarios
(1990~2013) with variables of hydrometeorological components for monthly drought forecast for
2006 and 2014, respectively. For drought forecasting to January 2006, for example, 16 historical
scenarios (1990~2000) of precipitation and temperature were inputted into hydrological models to
generate streamflows and groundwater level ensembles. For each forecasting period, the hydrological
model was executed with the hydrometeorological variables for the preceding 12 months to determine
the initial conditions. The historical data of each hydrometeorological component were then fitted to
their proper probability distribution to make the variable dimensionless. These ensembles finally
served as inputs in the calculation of the values of KSWSI with their weights. Fig. 6 shows the
procedure of monthly probabilistic drought forecasts.
[Fig. 6. Example of the procedure of the monthly probabilistic drought forecast]
*Calibration of the hydrological model: abcd water balance model*
In this study, the *abcd* water balance model was used, which has parameters of *a*, *b*, *c*, and *d* to
determine the streamflow and groundwater. The parameters of the *abcd* model are estimated with a

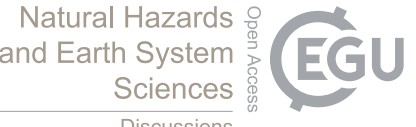



regional regression for ungauged basins because streamflow is gauged only at Yongdam and
Daecheng Dam. The regional regression equation was then formulated using the relationship between
each of the calibrated parameters and the site specific basin characteristics such as basin length,
drainage area, basin annual average precipitation, basin annual average potential evapotranspiration,
basin average land height, basin average land slope, basin drainage density, basin average temperature,
basin monthly maximum precipitation, basin monthly maximum potential evapotranspiration,
drainage relief, soil type, and basin total stream length. The calibrated parameters, *a*, *b*, *c*, and *d* of the
*abcd* model were obtained using gauged stations in nine multipurpose dams in Korea. Table 4 shows
the regional regression equations over all of Korea as a result of a step-wise regression technique.
Using these equations with basin characteristics of an ungauged basin, *a*, *b*, *c*, and *d* can be computed
and consequently the streamflow of the basin can be computed from the calibrated *abcd* model.

13                    [Table 4 Regression equations for the *a*, *b*, *c*, and *d* parameters]

15       To verify the estimated parameters of the *abcd* model using the regional regression equations, the

*abcd* model was applied to generate the monthly inflows at Yongdam Dam from 2002 to 2004 (period
#1) and from 2010 to 2013 (period #2). The calculated values of the R-Bias, R-RMSE, and $R^2$ during
period #1 were -0.06, 35, and 0.92, respectively, and those during period #2 were 0.11, 0.55, and 0.91,
respectively, suggesting that the model parameters are accurately estimated.
3.2 Results of monthly drought forecasts

22       Fig. 7 showed the forecast results of monthly drought using MSWSI and KSWSI equations in

April and December of 2006 and 2014, respectively. Drought-intensities in the drought forecasts
using the KSWSIs were stronger than in the MSWSI, and the drought occurred widely throughout the
Geum River basin. While the MSWSI-based drought forecasts for April 2006 and 2014 predicted
slight and moderate drought in some sub-basins of downstream and near Yongdam Dam, the results



of the KSWSI forecasted severe and moderate droughts in most sub-basins of the Geum River basin.
Then, in December 2006 and 2014, drought forecasts of MSWSIs were similar to those of KSWSIs;
especially, in December 2014, drought forecasts by KSWSIs showed severe droughts in some sub-
basins of downstream and near Yongdam Dam. Table 5 shows the occurrence probabilities of
droughts for each sub-basin by drought forecast using MSWSIs and KSWSI for April and December
2014. From the drought forecasts using KSWSI, the probabilities of severe droughts in both April and
December 2014 were over 70%, showing droughts were highly likely to occur.
In this study, the accuracy of the probabilistic forecast was measured using the Average Hit Score
(AHS) and Half Brier Score (HBS) (Wilks, 1995). The AHS scored the probabilities of occurrences of
drought forecasts for the drought category by the actual drought, and the ensemble drought forecasts
can be considered to be effective if their AHS is higher than the AHS of the naive forecasts. The
concept of HBS is similar to the mean square error and is a way to give a high score when ensemble
drought forecasts match the actual drought, but gives a penalty for wrong categories. The drought
forecast becomes increasingly more accurate as the HBS becomes smaller than the naive forecast. The
equations of AHS and HBS are as follows:
$$\text{AHS} = \frac{1}{N}\sum_{i=1}^{N} f_i^{\,o} \tag{3}$$
$$\text{HBS} = \frac{1}{N}\sum_{j=1}^{J}\sum_{i=1}^{N}(f_{i,j} - o_{i,j})^2 \tag{4}$$
where $f^{\,o}$ is the probability of drought forecast for the category of actual drought, $N$ is the number of
drought forecasts, $J$ is the number of drought categories, $f_{i,j}$ is the probability of the $i$th forecast in the
$j$th category, and $o_{i,j}$ is the actual drought in the $j$th category. The category of actual drought score is 1
at the $i$th drought forecast and the scores of the remaining categories are zero.
The drought forecasts were compared to the corresponding observed event for a verification
period of 12 months in 2006 and 2014. As shown in Table 6(a), the AHS of the 2006 and 2014 events



are 0.201 and 0.200, respectively, which are higher than that of the naive forecast (=0.174). Especially,
the AHSs of drought forecasts using KSWSI are 0.249 and 0.325 for 2006 and 2014, respectively,
which is more accurate than the drought forecast using MSWSI. The overall accuracy of the drought
forecasts was better in the dry season (October to the following May) than in the flood season (from
July to September), and the accuracy of drought forecasts using KSWSIs was improved from 0.219 to
0.397 by AHS. As shown in Table 6(b), while the accuracy of drought forecasts using the MSWSI is
0.848 in 2006, which is smaller than that of the naive forecast (=0.857) for 2006 and 2014, the
accuracy of MSWSI in 2014 (=0.865) was low. The accuracy of drought forecasts using KSWSI was
confirmed to be superior to that of the MSWSI because the HBS of K-SWSIs is 0.824 and 0.795 in
2006 and 2014, respectively. The occurrence ranges of actual forecasts and drought forecasts of
MSWSI and KSWSI were compared. Fig. 8(a) and 8(b) show the actual droughts (black dots) and
occurrence ranges of drought forecast ensembles (between the first and third quartiles of the box-plot)
from January to December 2014 for sub-basin 3001. The actual droughts exist in the range of the
drought forecast ensembles, implying that the drought forecasts consider the extent of the actual
drought and as the range of drought forecast ensembles narrows, including the occurrence of actual
drought, the accuracy of drought forecasts increases. While the ranges of drought forecasts using
MSWSI include most actual droughts in Fig. 8(a), the actual droughts are outside of the range of
drought forecasts of KSWSI in Fig. 8(b). As shown in Figs. 8(c) and 8(d) in sub-basin 3007, the
drought forecasts with MSWSI are effective because most categories of drought forecast ensembles
(red dashed boxes) include actual droughts. Especially, in Fig. 8(d), the box-plot is very small in the
drought forecasts, implying that the values of the KSWSI drought forecast ensemble are very
concentrated in the category of 'severely dry' and the actual drought also occurs in the same category,
so that the accuracy of the drought forecasts with KSWSI is very high. Fig. 8(e) and 8(f) show a
similar tendency to that of sub-basin 3007, confirming the high accuracy of the drought forecast using
the KSWSI.
[Fig. 7. Comparison of the drought forecasts using the MSWSI and KSWSI on April and December in



1                2006 and 2014]

[Table 5. Comparison of the most probable drought categories and their probabilities for each sub-

3               basin in April and December on 2014 year]

4            [Table 6. Accuracy of the MSWSI and KSWSI results]

[Fig. 8. Comparison of the drought forecasts ranges for each month at sub-basin 3001, 3007, and 3014

6                      in 2014 year]

4. Uncertainty Analysis of the Calculation Procedure for the KSWSI

9        In steps 1, 2, and 4 in the KSWSI calculation process described in Section 2.2, the researcher's

experience and subjective judgment are involved. For example, the researcher selects the
hydrometeorological variables as drought components and fits the probability distributions to the
selected drought components. This means that the final results using the KSWSI can differ according
to the researcher's subjective judgment; this likely results in uncertainty about the drought monitoring
and forecasts. The subjective judgments of the researchers for each stage of the KSWSI calculation
are as follows.
• Step 1&2: Analysis and selection of available hydrometeorological components for each basin
(a) selection of available hydrometeorological components
(b) data quality verification of selected hydrometeorological components
(c) selection of observation stations to acquire hydrometeorological data as drought

21        components

• Step 4: Estimation of probability distributions for each hydrometeorological component
(a) estimation of probability distributions for each drought component
(b) selection of proper probability distributions for each drought component



Therefore, in this section, the influence of researcher's subjective judgment on the KSWSI
calculation and its corresponding uncertainty are analyzed.
4.1 Analysis of the influence of expanding hydrometeorological components as KSWSI components
As mentioned above, in this study, the precipitation data, water level data, discharge data,
streamflow data, dam data (included in inflow, release, and storage data), and groundwater data were
selected as hydrometeorological components that can be practically applied as KSWSI drought
components. Table 7 shows that, for the MSWSI, observed data in only one station was used for each
drought component (K-water, 2005); however, averaged data were used from several stations in the
KSWSI calculation. Especially, in the case of precipitation, areal-averaged data using the Thiessen
method was used rather than point data. Secondly, only the data of Daecheong Dam was reflected in
MSWSI, because the data length of Yongdam Dam was insufficient at the time of the MSWSI study.
This study used the observation data of dams as follows: (1) for applying dam data, the sub-basins in
Geum River basin were divided into those that were affected by Yongdam Dam and those affected by
Daecheong Dam; (2) sub-basins around dams were also divided into upstream and downstream sub-
basins, and the observation data of dam inflow and storage in the upstream and dam release in
downstream were then applied to the KSWSI calculation, respectively. Finally, the MSWSI
calculation only reflected four drought components, but the K-SWSI reflected a maximum of six
drought components and the number of observation stations used to obtain meteorological data in all
drought components was increased.
In order to investigate the influence of the selection of hydrometeorological components, the
KSWSIs for 2001 and 2006 drought events were calculated using the drought components selected in
Table 2. Similar to the MSWSI studies (K-water, 2005), the probability distributions of all drought
components were assumed to be normal distributions. In Table 8, the results of both MSWSI and
KSWSI showed drought as a whole in all of the sub-basins. Especially, the same values of MSWSIs
were calculated from the same drought components from sub-basin 3001 to 3004, but KSWSIs show





slightly different drought indices. In the 2006 drought event, MSWSI indicated that the water
resources of the entire Geum River system were very low, resulting in drought. However, the opposite
results were recorded for the KSWSI, where drought was avoided due to the abundant water resources.
*Comparison of MSWSIs and KSWSIs in sub-basin 3001*
Fig. 9(a) shows the time series for the MSWSIs and the KSWSIs in sub-basin 3001 for the 2006
drought event. In both the MSWSIs and KSWSIs, drought occurred in the beginning of 2006, but the
drought was somewhat resolved as the flood season passed. However, the drought-intensity calculated
by KSWSIs is stronger than that by MSWSI. Fig. 9(b) shows the time-series of non-dimensional
ratios to the normal-year for the 2005-2006 years for each hydrometeorological component of
precipitation, streamflow, and dam inflow. In block A1 of Fig. 9(b) (same as block A of Fig. 9(a)), the
amount of precipitation and dam inflows were lower than the normal-year from January to April 2005,
and the streamflow was almost the same as normal-year. In block B1 of Fig. 9(b) (same as block B of
Fig. 9(a)), in July 2006, the dam inflow and streamflow both increased due to very large precipitation
compared to the normal-year, and since August, the dam inflow also decreased because precipitation
was very low. For the observed hydrometeorological data for March, June, and August 2006, while
the amount of streamflow is maintained, it is more reasonable that hydrological droughts occurred
because of the low precipitation and dam inflow.
*Comparison of MSWSIs and KSWSIs in sub-basin 3010*
Fig. 9(c) shows the time series for the MSWSIs and the KSWSIs in sub-basin 3010 for the 2006
drought event. While the results of MSWSIs show no drought in early 2006 but severe droughts in the
flood season, KSWSIs was below the category of 'near normal', except for July, and indicated that
water shortage occurred for the entire period. In block C1 of Fig. 9(d) (same as block C of Fig. 9(c)),
MSWSIs indicated that water resources were abundant, but some water shortages did actually occur,
and the accuracy of the KSWSI results is considered to be superior to that of MSWSI because
precipitation is very influential in this season. In block D1 of Fig. 9(d) (same as block D of Fig. 9(c)),




in July 2006, a large amount of precipitation occurred compared to the normal-year, so the amount of
both dam release and streamflow was increased and the water shortage was then resolved. After
August, the amounts of both dam release and streamflow decreased. The MSWSIs showed severe
drought in July when the amount of precipitation, streamflow, and dam release were larger than
normal-year, but the KSWSI results indicated that the drought was resolved. In 2006, the streamflow
and dam release were smaller than normal-year and their variation was not significant. Reflecting the
water resources, KSWSIs showed that droughts were resolved due to the occurrence of precipitation,
but water shortages had generally occurred.
As shown in the previous results, the results of KSWSIs may affect whether or not the actual
droughts are accurately simulated by the KSWSI calculation depending on the hydrometerological
components used as the drought components, which station data are obtained, and the length of used
data for each station, respectively.
[Table 7. Comparison of hydrometeorological components in each sub-basin between the MSWSI

15          and this KSWSI studies]

[Table 8. Comparison of MSWSI and KSWSI results in July in each sub-basin]
[Fig. 9. Verification of previous results and these MSWSI results in sub-basins 3001 and 3010: (a)

18          & (b) at 3001 and (c) & (d) at 3010]

4.2 Analysis of the influence of the selection of probability distributions for each KSWSI component
In Section 2.2, the precipitation component was fitted to the Gumbel and GEV distributions, the
normal and Gumbel distributions for streamflow, 2-parameter log-normal and Gumbel distributions
for dam data (inflow, release, and storage), and the 3-parameter log-normal distribution for
groundwater. Since the drought components which are applied for each sub-basin differ and several
probability distributions can be applied in the even same sub-basin, the KSWSI results can differ
depending on the probability distributions selected. In this study, we determined how the results could
be changed by calculating KSWSIs by applying all the probability distributions (including the normal





distribution) that are shown to be appropriate. Table 9 shows the probability distributions applied to
each drought component. In the application process, the maximum number of scenarios for
probability distributions applicable to the sub-basins is 36 (= 3 probability distributions for
precipitation × 2 for river flow × 3 for dam data × 2 for groundwater), and the ranges of KSWSIs are
indicated using the maximum and minimum values among these combinations (Fig. 10).

6       In Fig. 10(a), the maximum and minimum values of KSWSIs showed a similar tendency in the

2006 drought event, but the KSWSI ranges were separated by two to three categories. The time-series
of the maximum values of KSWSIs was located above the category 'near normal', which means
droughts did not occur, but the minimum values of KSWSIs showed droughts due to water shortage
except for July. The KSWSIs using only normal distribution are similar to the averages of the
maximum and minimum KSWSIs. In the 2014 drought event shown Fig. 10(b), the maximum values
of KSWSIs are also above the category 'near normal', which means the water resources are abundant
in 2014; however, the minimum KSWSIs shows continuous severe drought, similar to the results of
KSWSIs using only the normal distribution. The difference between the maximum and minimum
values of KSWSIs was significant (maximum five categories). In sub-basin 3008 shown Fig. 10(c),
the time-series of the maximum and minimum KSWSIs showed similar trends in the 2006 drought
event, and the ranges of the maximum and maximum KSWSIs differed by one to two categories.
Furthermore, most of the maximum KSWSIs did not show water shortages, and the minimum
KSWSIs showed droughts in March, August, and September. In Fig. 10(d) for the 2014 drought event,
while the maximum KSWSIs were almost similar to the minimums of them from January to May, the
maximum and minimum KSWSIs showed large differences in the flood season.

22       The scenario ranges of KSWSI generally varied according to the selection of probability

distributions, and their results of droughts significantly differed depending on the probability
distributions selected for each drought component. Therefore, it was confirmed that the selection of
the probability distributions could affect the accuracy of results of the KSWSI calculation.
[Table 9. Available probability distributions to each hydrometerological component at each sub-

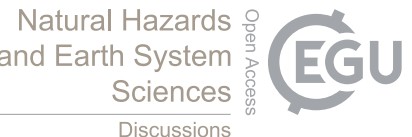

1        basin]

[Fig. 10. Comparison of MSWSI time series of max, min, and normal at sub-basin 3001 and 2008
in 2006 and 2014: (a) & (b) at 3001 & 3008, respectively, in 2006 and (c) & (d) at 3001 & 3008,

4        respectively, in 2014]

4.3 Quantification of Uncertainty in the KSWSI calculation procedure
*Methodology: Maximum entropy principle*

8        Shannon (1948) first introduced the use of entropy as a method to estimate uncertainty

quantitatively if the information context is obtained from probability distributions of a given set of
information. If probabilities of occurrences of a certain set of information are large, the amount of
information is small, and if their probabilities are small, the amount of information becomes large. If
$X$ is defined as a random variable with probability $p$, and $I(X)$ is the information context of $X$, entropy
$H(X)$ is given as follows:
$$H(X) = -\sum p_X(x)\ln p_X(x) = \sum p_X(x)I(X) = E[I(X)]$$    (5)

17       Maximum Entropy (ME) based on Shannon's entropy theory (1948) was proposed by Jaynes

(1957). When a certain set of information is given, based on the information, maximum entropy
theory provides the probability density function which maximizes the entropy. If a given set of
information is the minimum value $a$ and maximum value $b$, the distribution maximizing the entropy is
a uniform distribution on [a, b], and the corresponding entropy $H(X)$ (i.e. maximum entropy) is given
as (Gay and Estrada, 2010):
$$H(X) = -\int_a^b f_X(x)\ln f_X(x)dx = -\int_a^b \frac{1}{b-a}\ln\frac{1}{b-a}dx = -\ln(b-a)$$    (6)
*Uncertainty quantification of KSWSIs*



In this section, KSWSIs calculated by selected drought components and their probability
distributions in Section 4.2 are inputted into the formula (Eq. (6)) of ME to estimate and analyze
uncertainties of KSWSIs. The results are shown in Table 10 and Fig. 11. Of the ME values for each
sub-basin in Table 10(a), the ME value (=1.002) in sub-basin 3001 is the largest and the minimum
ME is 0.521 in sub-basin 3006 in the 2001 drought event. In 2006 and 2014, sub-basins 3002 and
3001 have the largest values of MEs of 1.120 and 1.503, respectively, and the smallest MEs of 0.578
and 0.578, respectively, in sub-basin 3012. Especially, even though the ME values of each sub-basin
slightly differ, MEs showed a similar tendency in the same sub-basin despite different drought events.
This tendency is more evident in the comparison of the number of ME values for each drought event,
drought component, and number of selected drought components for each sub-basin. In other words,
the ME values of the sub-basins with many drought components are large, and sub-basins with few
drought components, have relatively small ME values. The different drought components for each
sub-basin include the data of dam inflow, dam release, groundwater, and data of precipitation and
streamflow components, and were used in all sub-basins. Because the data of different observation
stations was used for each sub-basin, it could not be determined whether the difference of ME values
for each sub-basin was more influenced by dam and groundwater components than by precipitation
and streamflow. From the above results, it can be deduced that the increased number of drought
components does not necessarily improve the accuracy of the KSWSIs calculation to the actual
droughts. In other words, the large values of MEs imply that the results of KSWS have large
uncertainty. Therefore, only drought components that can represent the hydrometeorological
characteristic of each sub-basin should be selected and applied.

22       In the monthly MEs for each drought event in Table 10(b), the ME values (1.215 and 1.379) in

July are the largest and the minimum ME at 0.562 and 0.650 in January in the 2001 and 2006 drought
events, respectively. In 2014, the seasonal ME value was the highest at 1.053 in the flood season.
Furthermore, in all drought events, although the values of MEs decreased in the dry season, they
increased in the flood season as shown in Fig 11(b). To determine the reasons for this result, the
standard deviations of KSWSI according to the selected probability distributions in Section 2.3 are





also shown in Fig. 11(b). The trend of standard deviations of KSWSI was similar to the monthly MEs
for each drought event, which decreased in the dry season and increased in the flood season. The large
standard deviations of KSWSIs mean that the variation of calculated KSWSIs depending on the
selection of probability distributions is large, which affects the uncertainty of the KSWSI results. In
other words, the large MEs and standard deviations of KSWSIs in the flood season cause uncertainties,
which mean that the selection of the appropriate probability distributions for selected drought
components in the flood season is very important.

9       [Table 10. Maximum entropy results for each sub-basin and month in each drought event]

[Fig. 4. Comparison of maximum entropy results between sub-basins and months for each drought

11                 event]

5. Conclusion
In this study, the new hydrological drought index, KSWSI, was proposed, which overcomes
some of the limitations in the calculation of MSWSI applied in Korea. The probabilistic drought
forecasts based on ensemble technique were also conducted using KSWSI. The summary of the study
is as follows. Firstly, all hydrometeorological variables in the Geum River basin were investigated
and then classified into nine types. Based on these results, appropriate variables were selected as
drought components for each sub-basin. It was confirmed that the effect of precipitation component is
greater in the flood season. Secondly, to overcome the limitation of MSWSI, whereby only the normal
distributions are applied to all drought components, probability distributions suitable for each
hydrometeorological component were estimated. As a result of verifying the accuracy of KSWSIs
using historical observed meteorological data, the results of KSWSIs showed better drought
phenomenon in drought events. Thirdly, in this study, the monthly probabilistic drought forecasts
were calculated based on ensemble technique using KSWSI. The drought forecasts of both MSWSIs
and KSWSIs were more accurate than the naïve forecasts. In addition, in 2006 and 2014, both AHS
and HBS of the drought forecasts using KSWSIs were higher than those using MSWSI, demonstrating



that KSWSI is able to enhance the accuracy of drought forecasts. Finally, the influence of expanding
hydrometeorological components as KSWSI components was analyzed and the probability
distributions for each KSWSI component were selected. It is confirmed that the accuracy of KSWSIs
may be affected by the choice of hydrometerological components used as drought components, the
station data obtained, the length of used data for each station, and the probability distributions selected
for each drought component. Furthermore, the uncertainty quantification of the KSWSI calculation
procedure was also carried out. The large MEs and standard deviations of KSWSIs in the flood season
cause uncertainties, implying that the selection of the appropriate probability distributions for selected
drought components in the flood season is very important.
In order to monitor accurate droughts and manage water resources to mitigate droughts, in future
research, analysis will be needed not only of the spatially segmented sub-basin divisions, but also the
municipal district units in the administrative districts. This is because it is very important to
distinguish between the waterworks and the dam beneficiation regions and, for these regions, the
dams should be assessed individually by using the dam water supply capacity index. Further studies
should also be conducted on the practical use of meteorological forecasting data to improve the
accuracy of drought forecasts.
Acknowledgement
This study was funded by the Korea Meteorological Administration Research and Development
Program under Grant KMIPA-2015-6190

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





List of Tables
Table 1. Basic investigation of hydrometerological components at each sub-basin
Table 2. Selected hydrometerological components and stations at each sub-basin
Table 3. Selected suitable probability distributions to hydrometerological components at each sub-
basin
Table 4 Regression equations for the *a*, *b*, *c*, and *d* parameters
Table 5. Comparison of the most probable drought categories and their probabilities for each sub-
basin in April and December on 2014 year
Table 6. Accuracy of the MSWSI and KSWSI results
Table 7. Comparison of hydrometeorological components in each sub-basin between the MSWSI and
this KSWSI studies
Table 8. Comparison of MSWSI and KSWSI results in July in each sub-basin
Table 9. Available probability distributions to each hydrometerological component at each sub-basin
Table 10. Maximum entropy results for each sub-basin and month in each drought event





Table 1. Basic investigartion of hydrometerological components at each sub-basin
(a) Investigation of available hydrometerological components

| Basin No. | Sub-basin name | Pcp. station | WL station | W station | NS | LS level 1 | LS level 2 | WWS | LWS | GW |
|---|---|---|---|---|---|---|---|---|---|---|
| 3001 | Yongdam dam | O | O | O | X | O | O | O | O | O |
| 3002 | Downstream of Yongdam dam | X | O | X | X | O | O | X | X | O |
| 3003 | Muju Namdaecheon | O | O | X | X | O | O | X | O | O |
| 3004 | Youngdongcheon | O | O | O | X | O | O | X | O | O |
| 3005 | Chogang | O | O | O | X | O | O | X | O | O |
| 3006 | Upstream of Daecheong dam | O | O | X | O | O | O | X | O | O |
| 3007 | Bocheongcheon | O | O | O | O | O | O | X | O | O |
| 3008 | Daecheong dam | O | O | X | O | X | O | O | O | O |
| 3009 | Gapcheon | O | O | O | O | O | O | X | O | O |
| 3010 | Downstream of Daecheong dam | O | O | X | O | X | O | X | O | O |
| 3011 | Mihocheon | O | O | O | O | O | O | O | O | O |
| 3012 | Geum river Gongju | O | O | O | O | O | O | O | O | O |
| 3013 | Nonsancheon | O | O | X | O | O | O | X | O | O |
| 3014 | Geum river estuary bank | O | O | X | O | O | O | O | O | O |

* Pcp: Precipitation; WL: Water Level, W: Weather; NS: National Stream; WWS; Wide Water
Supply; LWS; Local Water Supply; GW: GroundWater
(b) Analysis and collection of hydrometerological components

| Components | Stations | Data length | Description |
|---|---|---|---|
| Precipitation | KMA: 9, MOLIT: 24, K-water: 8 | Maximum: 1966-2015 | . Data quality & length<br>. Priority to KMA<br>. Areal average with Thiessen method |
| Water level & streamflow | 87 | Maximum: 1990-2015 | . Data quality & length |
| Dam | Yongdam, Daecheong | Yongdam: 2001-2015 Daecheong: 1981-2015 | . Total nine dams located<br>. non-available 6 dams in KRC |
| Groundwater | 7 | Maximum: 1998-2015 | . Used in GIMS<br>. Data quality & length |



1    Table 2. Selected hydrometeorological components and stations at each sub-basin

| Basin No. | Subbasin classification | Hydrometeorological components | | | |
|---|---|---|---|---|---|
| | | Precipitation | Streamflow | Dam | Groundwater |
| 3001 | Upstream of dam | Jangsu, Daebul, Buksang, Jinan | Donghyang, Chunchun | Inflow & water-level in Yongdam dam | Jangsu-Jangsu |
| 3002 | Downstream of dam | Muju(KW) | Anchun | Release discharge in Yongdam dam | |
| 3003 | Precipitation, Streamflow | Muju(KW), Buksang, Muju(M) | Sulchun, Jangbaek | | |
| 3004 | Precipitation, Streamflow | Geumsan(K), Geumsan(KW), Youngdong | Sutong, Hotan | | Geumsan-Geumsan, Geumsan-Boksu |
| 3005 | Precipitation, Streamflow | Chupoongryung, Hwanggan, Buhang2 | Songchun, Simchun | | |
| 3006 | Precipitation, Streamflow | Iwon | Okchun | | |
| 3007 | Precipitation, Streamflow | Boeun(K), Boeun(KW), Neungwol | Gidaegyo, Chungsung | | |
| 3008 | Upstream of dam | Gunbuk, Annae | Okgakgyo, Daechung dam, Hyundo | Inflow & water-level in Daechung dam | |
| 3009 | Downstream of dam | Daecheon | Bangdong, Sindae | | Daejeon-Moonpyung, Daejeon-Taepyung |
| 3010 | Precipitation, Streamflow | Bugang | Bugang, Maepo | Release discharge in Daechung dam | |
| 3011 | Precipitation, Groundwater | Cheongju, Chunan, Gaduk, Sunghwan, Byungcheon, Jeungpyung, Jinchun, Oryu | Chungju, Hapgang, Mihogyo | | Chungwon-Gaduk, Jinchun-Jinchun |
| 3012 | Precipitation, Streamflow | Buyeo, Chungyang, Jungsan, Banpo, Bokryong, Gongju, Hongsan, Jungan | Guryong, Gyuam | | |
| 3013 | Precipitation, Streamflow | Yeonsan, Jangsun, Ganggyung | Hangwol, Nonsan | | |
| 3014 | Precipitation, Streamflow | Gunsan, Hamyeol, Ganggyung | Ippo, Okpo | | |

2    * KW: K-water; K: KMA; M: MLIT





1    Table 3. Selected suitable probability distributions to hydrometerological components at each sub-

2    basin

| Basin No. | Hydrometeorological components | | | |
|---|---|---|---|---|
| | Precipitation | Streamflow | Dam | Groundwater |
| 3001 | Gumbel | Gumbel | 2-Log-Normal | 3-Log-Normal |
| 3002 | Gumbel | Normal | 2-Log-Normal | |
| 3003 | Gumbel | Normal | | |
| 3004 | Gumbel | Gumbel | | 3-Log-Normal |
| 3005 | Gumbel | Gumbel | | |
| 3006 | Gumbel | Gumbel | | |
| 3007 | Gumbel | Gumbel | | |
| 3008 | Gumbel | Gumbel | 2-Log-Normal | |
| 3009 | Gumbel | Normal | | 3-Log-Normal |
| 3010 | Gumbel | Gumbel | 2-Log-Normal | |
| 3011 | Gumbel | Gumbel | | 3-Log-Normal |
| 3012 | Gumbel | Gumbel | | |
| 3013 | Gumbel | Gumbel | | |
| 3014 | Gumbel | Gumbel | | |





1    Table 4. Regression equations for the $a$, $b$, $c$, and $d$ parameters

| | Regression equations |
|---|---|
| $a$ | = 0.1472 - 0.6002×(basin average temperature) + 0.01236×(basin annual average potential evapotranspiration) - 0.0602×(basin drainage density) |
| $b$ | = -895.3440 + 1.0696×(basin annual average potential evapotranspiration) + 256.8310 × (basin drainage density) + 1.3901×(basin monthly maximum precipitation) + 0.0789×(basin total stream length) |
| $c$ | = -0.3893 + 0.9773×(basin average temperature) + 0.0196× (basin annual average potential evapotranspiration) - 0.10182×(basin drainage density) - 0.0006×(basin monthly maximum precipitation) |
| $d$ | = -3.7841 + 0.0128×(basin annual average potential evapotranspiration) + 0.0427×(basin annual average precipitation) + 0.3206×(basin drainage density) |




1   Table 5. Comparison of the most probable drought categories and their probabilities for each sub-

2   basin in April and December on 2014 year

| Basin No. | With the MSWSI | | | | With the KSWSI | | | |
|---|---|---|---|---|---|---|---|---|
| | April 2014 | | December 2014 | | April 2014 | | December 2014 | |
| | Category | Prob. | Category | Prob. | Category | Prob. | Category | Prob. |
| 3001 | 4 | 41.9 | 3 | 32.3 | 7 | 32.3 | 4 | 48.4 |
| 3002 | 4 | 48.4 | 7 | 22.6 | 4 | 35.5 | 7 | 29.0 |
| 3003 | 6 | 32.3 | 6 | 32.3 | 7 | 38.7 | 6 | 41.9 |
| 3004 | 4 | 64.5 | 3 | 38.7 | 5 | 51.6 | 4 | 51.6 |
| 3005 | 6 | 25.8 | 4 | 29.0 | 7 | 77.4 | 7 | 77.4 |
| 3006 | 4 | 32.3 | 6 | 32.3 | 7 | 77.4 | 7 | 77.4 |
| 3007 | 6 | 25.8 | 4 | 25.8 | 7 | 77.4 | 7 | 77.4 |
| 3008 | 4 | 67.7 | 4 | 71.0 | 5 | 35.5 | 6 | 35.5 |
| 3009 | 4 | 54.8 | 3 | 45.2 | 4 | 54.8 | 3 | 51.6 |
| 3010 | 4 | 74.2 | 4 | 64.5 | 7 | 38.7 | 5 | 29.0 |
| 3011 | 4 | 51.6 | 3 | 32.3 | 5 | 51.6 | 4 | 54.8 |
| 3012 | 6 | 29.0 | 4 | 25.8 | 7 | 77.4 | 7 | 77.4 |
| 3013 | 6 | 32.3 | 7 | 41.9 | 7 | 77.4 | 7 | 77.4 |
| 3014 | 5 | 32.3 | 7 | 25.8 | 7 | 77.4 | 7 | 77.4 |



1    Table 6. Accuracy of the MSWSI and KSWSI results

2    (a) Average Hit Score

| Month | MSWSI | | KSWSI | | Season | MSWSI | | KSWSI | |
|---|---|---|---|---|---|---|---|---|---|
| | 2006 | 2014 | 2006 | 2014 | | 2006 | 2014 | 2006 | 2014 |
| 1 | 0.230 | 0.212 | 0.348 | 0.491 | Spring | 0.197 | 0.235 | 0.195 | 0.314 |
| 2 | 0.273 | 0.260 | 0.342 | 0.507 | | | | | |
| 3 | 0.093 | 0.240 | 0.354 | 0.182 | | | | | |
| 4 | 0.258 | 0.309 | 0.096 | 0.369 | | | | | |
| 5 | 0.239 | 0.157 | 0.134 | 0.392 | Summer | 0.168 | 0.184 | 0.213 | 0.354 |
| 6 | 0.224 | 0.242 | 0.177 | 0.332 | | | | | |
| 7 | 0.099 | 0.129 | 0.075 | 0.459 | | | | | |
| 8 | 0.180 | 0.182 | 0.388 | 0.272 | Autumn | 0.214 | 0.167 | 0.248 | 0.176 |
| 9 | 0.199 | 0.141 | 0.360 | 0.237 | | | | | |
| 10 | 0.252 | 0.210 | 0.286 | 0.104 | | | | | |
| 11 | 0.193 | 0.152 | 0.099 | 0.187 | Winter | 0.225 | 0.214 | 0.340 | 0.455 |
| 12 | 0.171 | 0.171 | 0.329 | 0.366 | | | | | |
| Average | 0.201 | 0.2 | 0.249 | 0.325 | | | | | |



1    (b) Half Brier Score

| Month | MSWSI | | KSWSI | | Season | MSWSI | | KSWSI | |
|---|---|---|---|---|---|---|---|---|---|
| | 2006 | 2014 | 2006 | 2014 | | 2006 | 2014 | 2006 | 2014 |
| 1 | 0.851 | 0.840 | 0.694 | 0.494 | Spring | 0.844 | 0.805 | 0.963 | 0.801 |
| 2 | 0.730 | 0.761 | 0.627 | 0.442 | | | | | |
| 3 | 1.059 | 0.805 | 0.665 | 1.081 | | | | | |
| 4 | 0.724 | 0.680 | 1.133 | 0.693 | | | | | |
| 5 | 0.748 | 0.931 | 1.090 | 0.630 | Summer | 0.889 | 0.872 | 0.918 | 0.754 |
| 6 | 0.768 | 0.755 | 0.961 | 0.780 | | | | | |
| 7 | 1.023 | 0.977 | 1.180 | 0.554 | | | | | |
| 8 | 0.878 | 0.885 | 0.613 | 0.929 | Autumn | 0.833 | 0.944 | 0.772 | 1.079 |
| 9 | 0.853 | 0.969 | 0.638 | 0.937 | | | | | |
| 10 | 0.789 | 0.899 | 0.792 | 1.232 | | | | | |
| 11 | 0.857 | 0.962 | 0.886 | 1.067 | Winter | 0.824 | 0.837 | 0.645 | 0.545 |
| 12 | 0.891 | 0.910 | 0.613 | 0.698 | | | | | |
| Average | 0.848 | 0.865 | 0.824 | 0.795 | | | | | |





Table 7. Comparison of hydrometeorological components in each sub-basin between the MSWSI and
this KSWSI studies

| Basin No. | MSWSI Study | KSWSI study | Subbasin classification |
|-----------|-------------|-------------|-------------------------|
| 3001 | D_DF., SF(1 OB), Pcp(1 OB) | Y_DF & Y_DWL, SF(2 OBs), Pcp(4 OBs), GW(1 OB) | Upstream of dam |
| 3002 | D_DF, SF(1 OB), Pcp(1 OB) | Y_DRD, SF(2 OBs), Pcp(4 OBs) | Downstream of dam |
| 3003 | D_DF, SF(1 OB), Pcp(1 OB) | SF(2 OBs), Pcp(3 OBs) | Precipitation, Streamflow |
| 3004 | D_DF, SF(1 OB), Pcp(1 OB) | SF(3 OBs), Pcp(2 OBs), GW(2 OBs) | Precipitation, Streamflow |
| 3005 | D_DF, SF(1 OB), Pcp(1 OB) | SF(2 OBs), Pcp(3 OBs) | Precipitation, Streamflow |
| 3006 | D_DF, SF(1 OB), Pcp(1 OB) | SF(1 OB), Pcp(1 OB) | Precipitation, Streamflow |
| 3007 | D_DF, SF(1 OB), Pcp(1 OB) | SF(2 OBs), Pcp(3 OBs) | Precipitation, Streamflow |
| 3008 | D_DF, Pcp(1 OB) | D_DF & D_DWL, SF(3 OBs), Pcp(2 OBs) | Upstream of dam |
| 3009 | SF(1 OB), Pcp(1 OB), GW(1 OB) | SF(2 OBs), Pcp(1 OB), GW(2 OBs) | Downstream of dam |
| 3010 | Pcp(1 OB) | D_DRD, SF(2 OBs), Pcp(1 OB) | Precipitation, Streamflow |
| 3011 | SF(1 OB), Pcp(1 OB), GW(1 OB) | SF(3 OBs), Pcp(8 OBs), GW(2 OBs) | Precipitation, Groundwater |
| 3012 | SF(1 OB), Pcp(1 OB) | SF(2 OBs), Pcp(8 OBs) | Precipitation, Streamflow |
| 3013 | Pcp(1 OB) | SF(2 OBs), Pcp(3 OBs) | Precipitation, Streamflow |
| 3014 | Pcp(1 OB) | SF(2 OBs), Pcp(3 OBs) | Precipitation, Streamflow |

* Y_: Yongdam dam, D_: Daecheong dam, DF: Dam Inflow, DWL: Dam WaterLevel, DRD: Dam
Release Discharge, Pcp: Precipitation, SF: StreamFlow, WL: WaterLevel, GW: GroundWater, OB:
Observed station





2    Table 8. Comparison of MSWSI and KSWSI results in July in each sub-basin

| Basin No. | MSWSI result (category) | | KSWSI result (category) | |
|---|---|---|---|---|
| | 2001 | 2006 | 2001 | 2006 |
| 3001 | -1.95(5) | 2.91(2) | -0.49(4) | 3.48(1) |
| 3002 | -1.95(5) | 2.91(2) | -0.41(4) | 0.98(4) |
| 3003 | -1.95(5) | 2.91(2) | -2.08(6) | 4.03(1) |
| 3004 | -1.95(5) | 2.91(2) | -1.09(5) | 3.68(1) |
| 3005 | -2.76(6) | 0.739(4) | 0.87(4) | 3.80(1) |
| 3006 | **-0.91(4)** | 2.01(2) | -2.46(6) | 3.74(1) |
| 3007 | -2.66(6) | 1.45(3) | -3.55(7) | 3.50(1) |
| 3008 | -2.80(6) | 2.69(2) | -2.47(6) | 3.69(1) |
| 3009 | -3.16(7) | 1.89(3) | -3.21(7) | 1.41(3) |
| 3010 | -2.49(6) | 2.39(2) | -2.41(6) | 3.36(1) |
| 3011 | -2.14(6) | 1.65(3) | -1.94(5) | 3.35(1) |
| 3012 | 0.53(4) | 0.40(4) | -1.76(5) | 2.51(2) |
| 3013 | -1.45(5) | 2.70(2) | -3.20(7) | 3.49(1) |
| 3014 | -0.77(4) | 2.70(2) | -1.92(5) | 3.23(1) |





2    Table 9. Available probability distributions to each hydrometerological component at each sub-basin

| Basin No. | KSWSI components | | | |
|-----------|------------------|------------|-----------------|-------------|
|           | Precipitation | Streamflow | (related to) Dam | Groundwater |
| 3001 | ·Gumbel<br>·GEV<br>·Normal | ·Gumbel<br>·Normal | ·2-Log-Normal<br>·Gumbel<br>·Normal | ·3-Log-Normal<br>·Normal |
| 3002 | ·Gumbel<br>·GEV<br>·Normal | ·Gumbel<br>·Normal | ·2-Log-Normal<br>·Gumbel<br>·Normal | |
| 3003 | ·Gumbel<br>·GEV<br>·Normal | ·Gumbel<br>·Normal | | |
| 3004 | ·Gumbel<br>·GEV<br>·Normal | ·Gumbel<br>·Normal | | ·3-Log-Normal<br>·Normal |
| 3005 | ·Gumbel<br>·GEV<br>·Normal | ·Gumbel<br>·Normal | | |
| 3006 | ·Gumbel<br>·GEV<br>·Normal | ·Gumbel<br>·Normal | | |
| 3007 | ·Gumbel<br>·GEV<br>·Normal | ·Gumbel<br>·Normal | | |
| 3008 | ·Gumbel<br>·GEV<br>·Normal | ·Gumbel<br>·Normal | ·2-Log-Normal<br>·Gumbel<br>·Normal | |
| 3009 | ·Gumbel<br>·GEV<br>·Normal | ·Gumbel<br>·Normal | | ·3-Log-Normal<br>·Normal |
| 3010 | ·Gumbel<br>·GEV<br>·Normal | ·Gumbel<br>·Normal | ·2-Log-Normal<br>·Gumbel<br>·Normal | |
| 3011 | ·Gumbel<br>·GEV<br>·Normal | ·Gumbel<br>·Normal | | ·3-Log-Normal<br>·Normal |
| 3012 | ·Gumbel<br>·GEV<br>·Normal | ·Gumbel<br>·Normal | | |
| 3013 | ·Gumbel<br>·GEV<br>·Normal | ·Gumbel<br>·Normal | | |
| 3014 | ·Gumbel<br>·GEV<br>·Normal | ·Gumbel<br>·Normal | | |





2      Table 10. Maximum entropy results for each sub-basin and month in each drought event

4      (a) For each sub-basin

| Basin No. | Maximum entropy | | | Average |
|---|---|---|---|---|
| | 2001 | 2006 | 2014 | |
| 3001 | 1.002 | 1.198 | 1.503 | 1.234 |
| 3002 | 0.985 | 1.210 | 1.352 | 1.182 |
| 3003 | 0.845 | 0.785 | 0.985 | 0.872 |
| 3004 | 0.985 | 1.002 | 1.052 | 1.013 |
| 3005 | 0.789 | 0.812 | 1.005 | 0.869 |
| 3006 | 0.521 | 0.651 | 0.785 | 0.652 |
| 3007 | 0.742 | 0.584 | 0.712 | 0.679 |
| 3008 | 0.854 | 0.888 | 0.616 | 0.786 |
| 3009 | 0.795 | 0.875 | 0.687 | 0.786 |
| 3010 | 0.891 | 0.985 | 0.871 | 0.916 |
| 3011 | 0.841 | 0.784 | 0.852 | 0.826 |
| 3012 | 0.668 | 0.578 | 0.363 | 0.537 |
| 3013 | 0.784 | 0.652 | 0.514 | 0.650 |
| 3014 | 0.781 | 0.587 | 0.612 | 0.660 |

6      (b) For each month

| Month | Maximum entropy | | | Average | Season | Averaged ME |
|---|---|---|---|---|---|---|
| | 2001 | 2006 | 2014 | | | |
| 1 | 0.562 | 0.650 | 0.541 | 0.584 | Spring | 0.787 |
| 2 | 0.701 | 0.716 | 0.629 | 0.682 | | |
| 3 | 0.825 | 0.765 | 0.882 | 0.824 | | |
| 4 | 0.795 | 0.827 | 0.722 | 0.781 | Summer | 1.053 |
| 5 | 0.721 | 0.847 | 0.697 | 0.755 | | |
| 6 | 0.854 | 0.785 | 0.865 | 0.835 | | |
| 7 | 1.215 | 1.379 | 1.174 | 1.256 | Autumn | 0.904 |
| 8 | 1.125 | 1.087 | 0.992 | 1.068 | | |
| 9 | 0.987 | 1.182 | 1.077 | 1.082 | | |
| 10 | 1.002 | 0.843 | 0.883 | 0.909 | Winter | 0.676 |
| 11 | 0.785 | 0.686 | 0.695 | 0.722 | | |
| 12 | 0.625 | 0.889 | 0.768 | 0.761 | | |



3                                 List of Figures






2
3

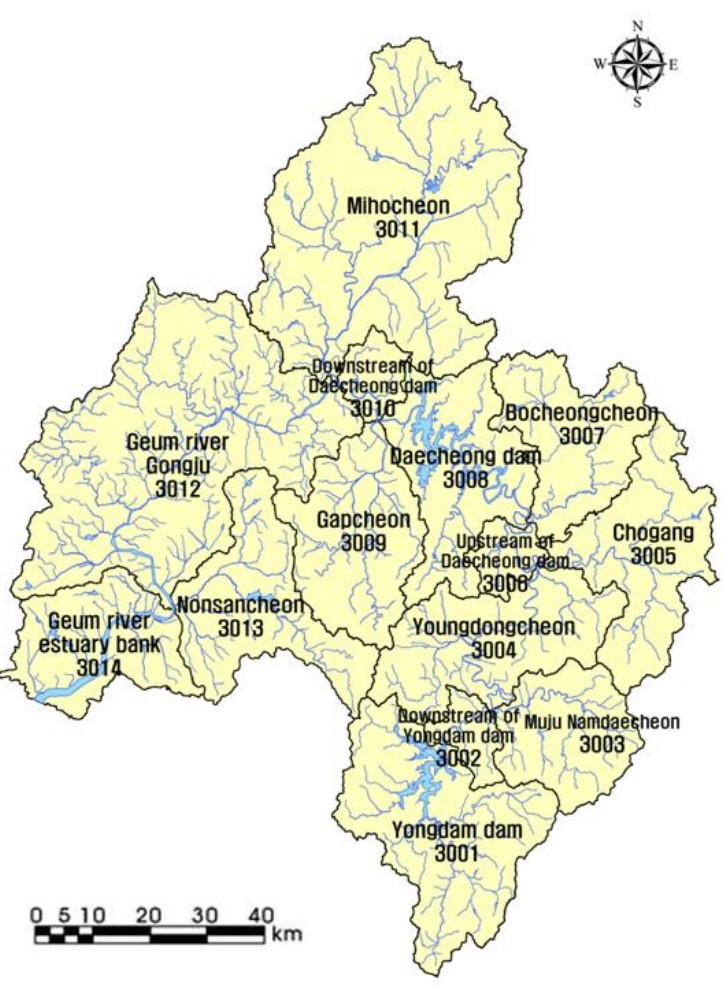

6    Fig. 1. Study basin: 14 sub-basins in Geum River basin




2    Fig. 2. Procedure of the previous MSWSI calculation and two improvements proposed by this study






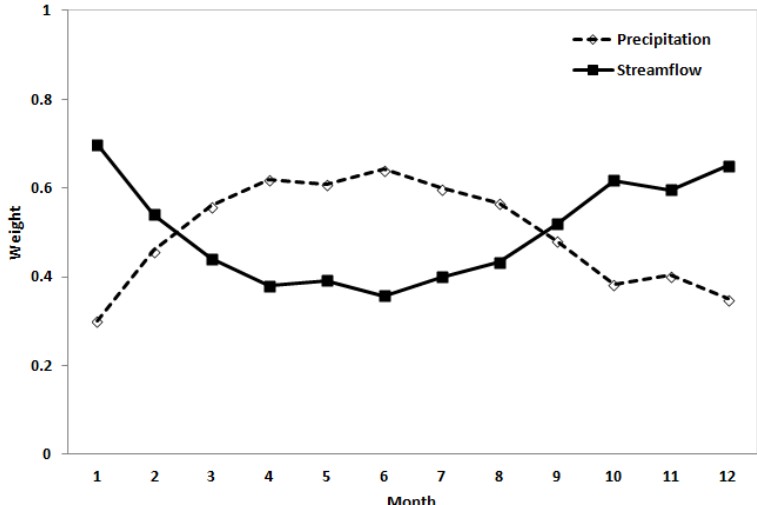

3               (a) Sub-basin 3001

6               (b) Sub-basin 3007

8     Fig. 3. Example of weights of each drought component for each month at sub-basin 3001 and 3007



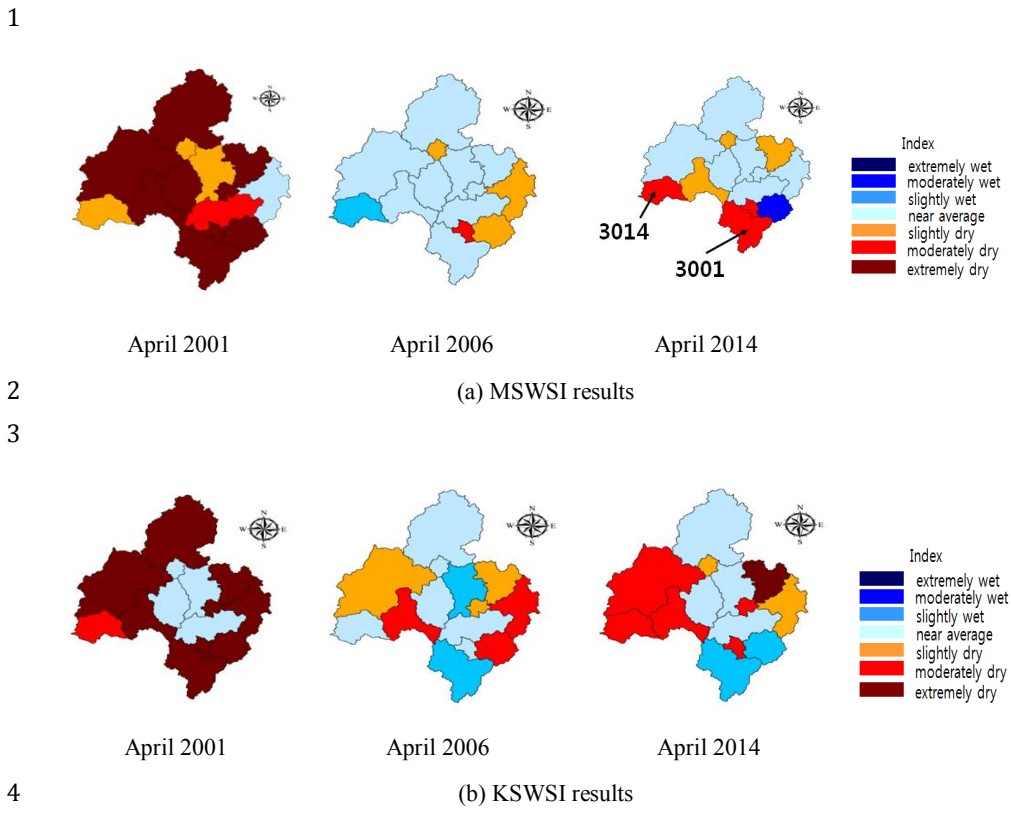

2                                     (a) MSWSI results

4                                     (b) KSWSI results

6   Fig. 4. Comparison of the MSWSI and KSWSI results in April 2001, 2006, and 2014 years

17




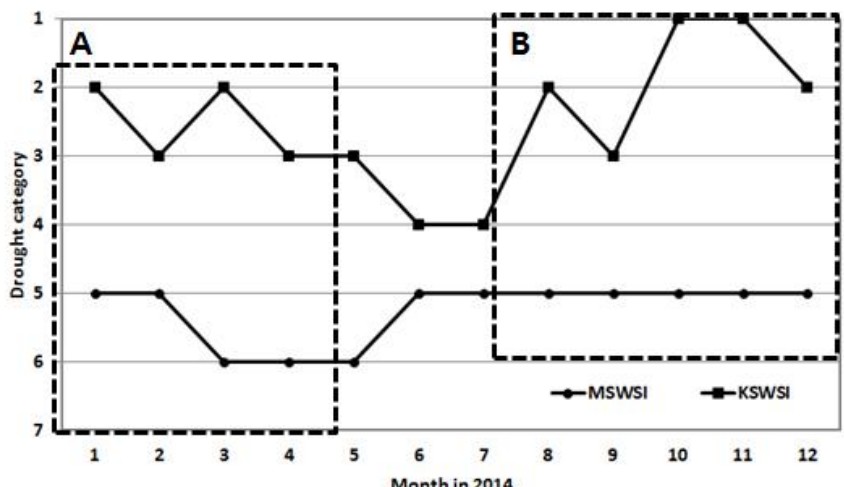

3                    (a) Time series of the previous and improved monthly MSWSIs in 2014 year

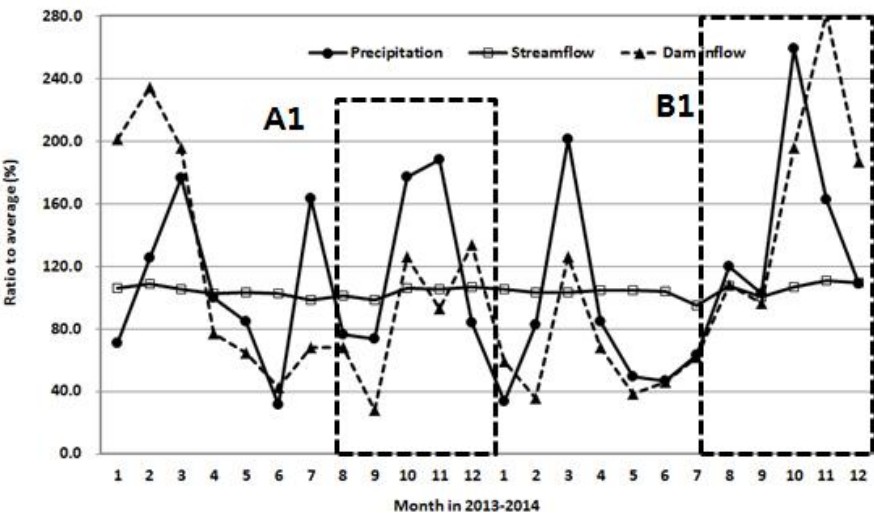

6              (b) Time series of monthly precipitation, water level, and dam inflow in 2014 year




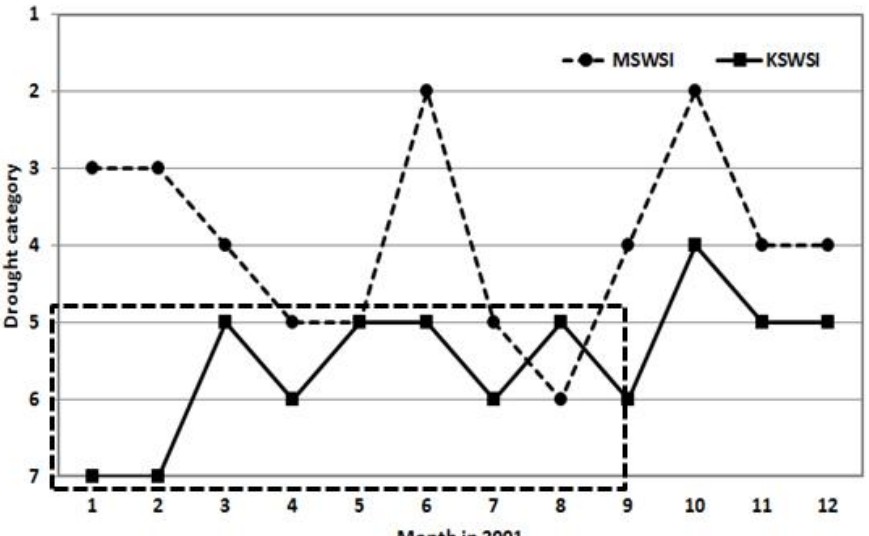

2          (c) Time series of the previous and improved monthly MSWSIs in 2001 year

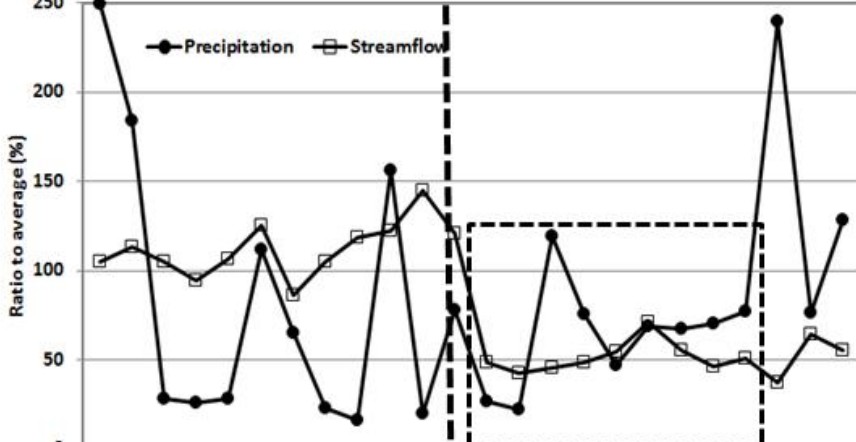

5          (d) Time series of monthly precipitation and streamflow in 2001 year

7     Fig. 5. Verification of improved MSWSI in sub-basin 3001 and 3014: (a) & (b) at 3001 and (c) & (d)

8     at 3014





Fig. 6. Example of the procedure of the monthly probabilistic drought forecast



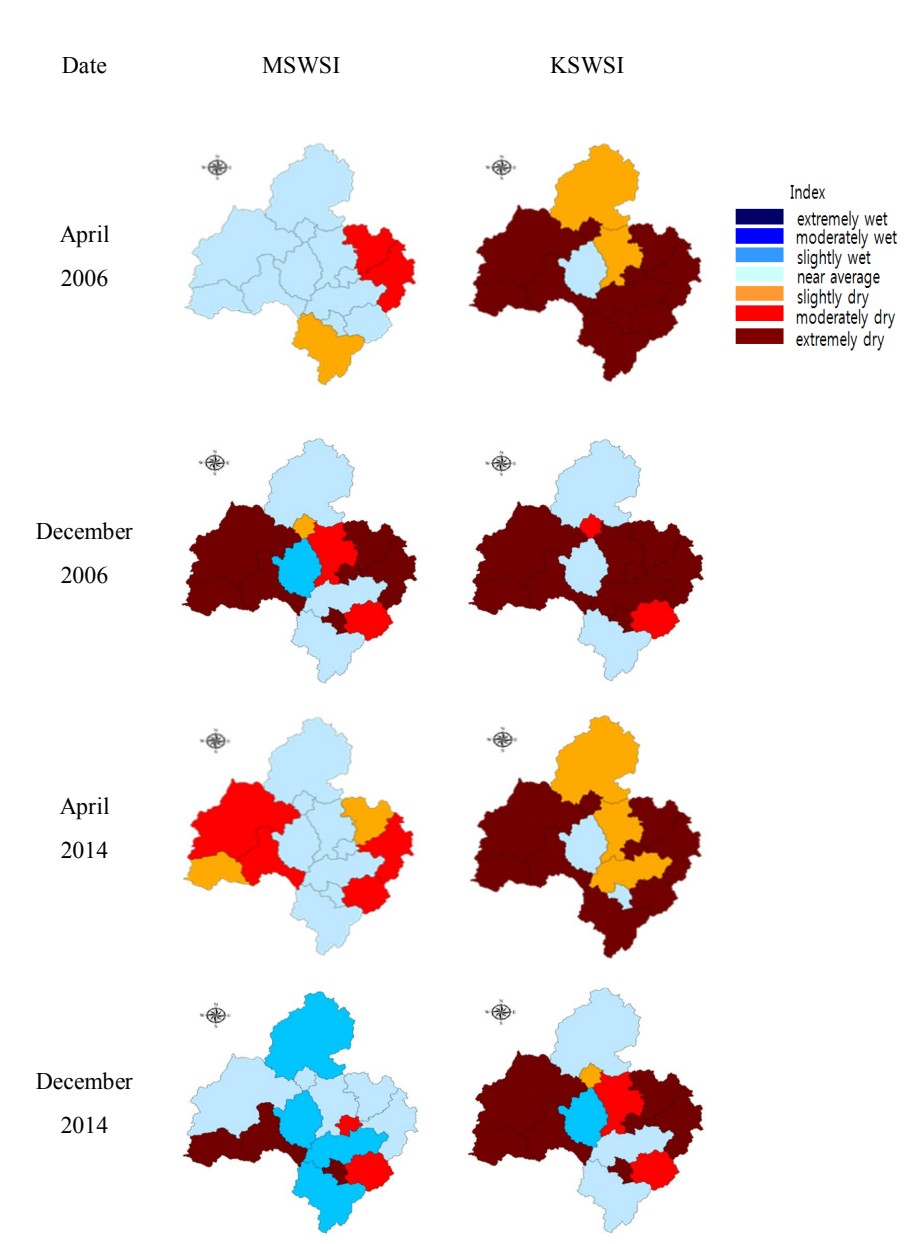

3    Fig. 7. Comparison of the drought forecasts using the MSWSI and KSWSI on April and December in

4    2006 and 2014 years





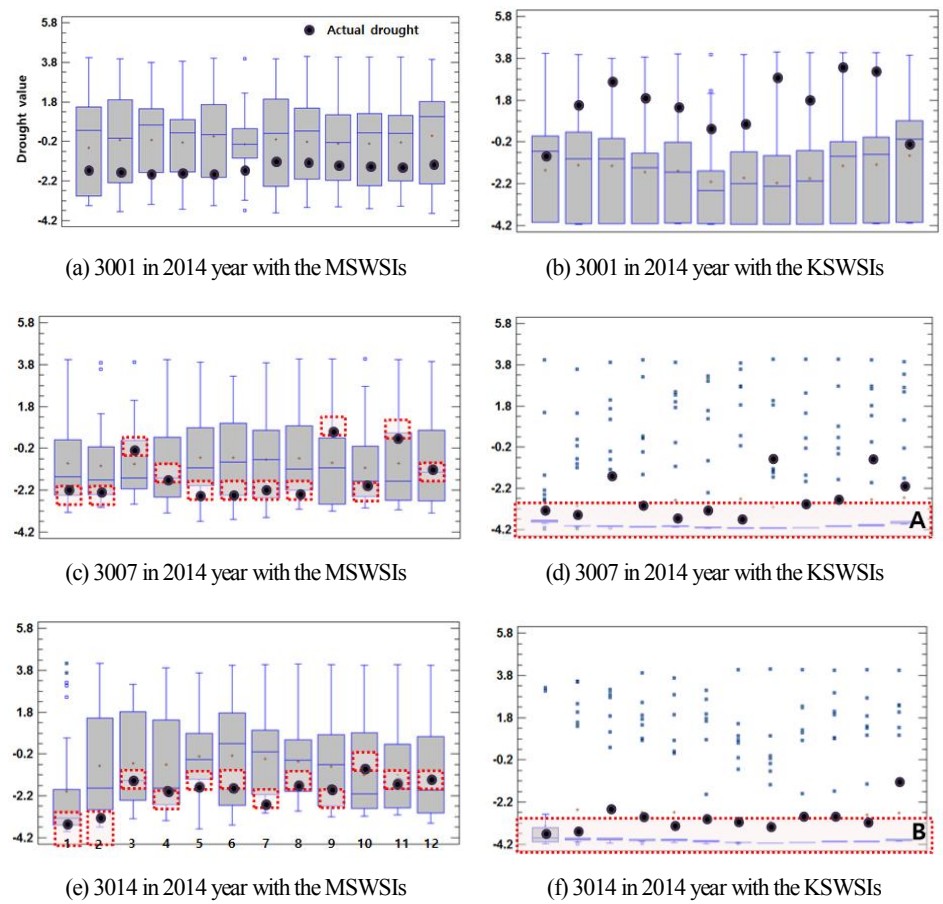

Fig. 8. Comparison of the drought forecasts ranges for each month at sub-basin 3001, 3007, and 3014
in 2014 year





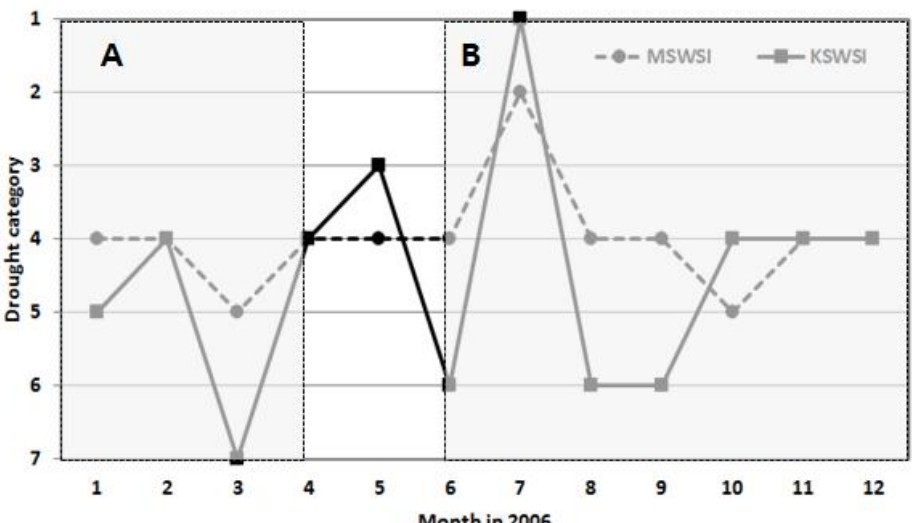

(a) Time series of the previous and this monthly MSWSI results in 2006 year at 3001

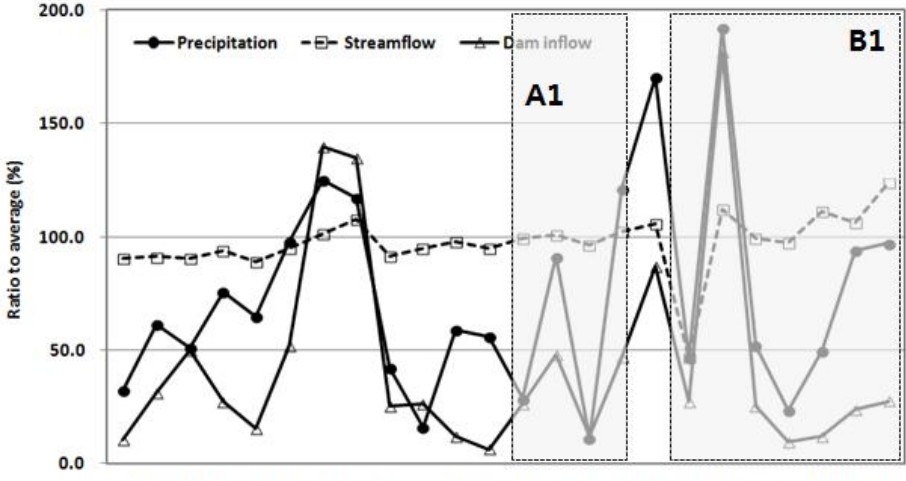

(b) Time series of monthly precipitation, streamflow, and dam inflow in 2005-2006 years at 3001





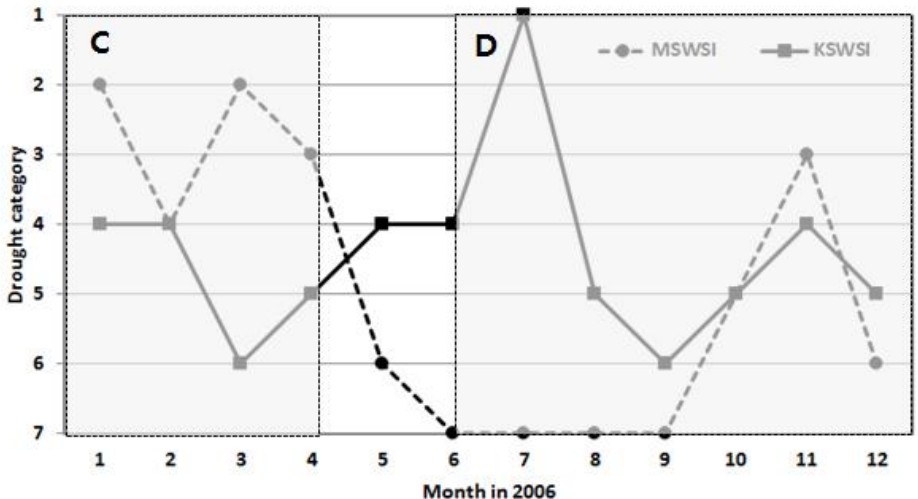

2          (c) Time series of the previous and this monthly MSWSI results in 2006 year at 3010

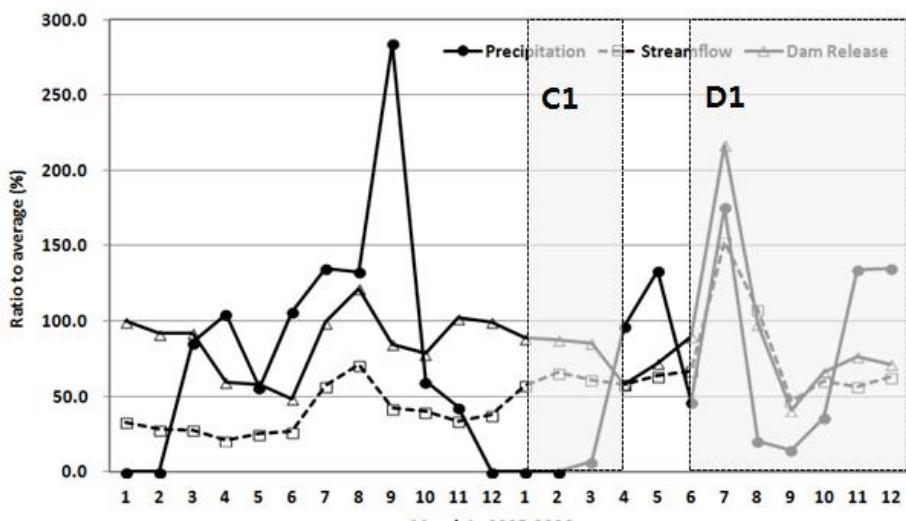

5          (d) Time series of monthly precipitation, streamflow, and dam release in 2005-2006 years at 3010

Fig. 9. Verification of MSWSI and KSWSI results in sub-basins 3001 and 3010: (a) & (b) at 3001 and
(c) & (d) at 3010



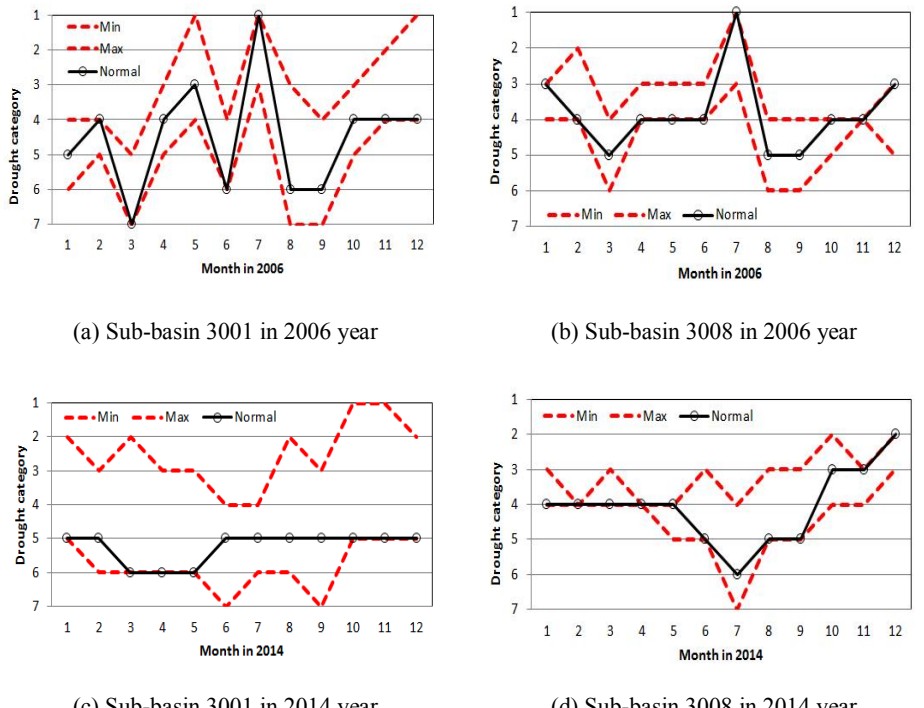

(a) Sub-basin 3001 in 2006 year      (b) Sub-basin 3008 in 2006 year

(c) Sub-basin 3001 in 2014 year      (d) Sub-basin 3008 in 2014 year

1    Fig. 10. Comparison of KSWSI time series of max, min, and normal at sub-basin 3001 and 3008 in

2    2006 and 2014 years: (a) & (b) at 3001 & 3008, respectively, in 2006 year and (c) & (d) at 3001 &

3    3008, respectively, in 2014 year




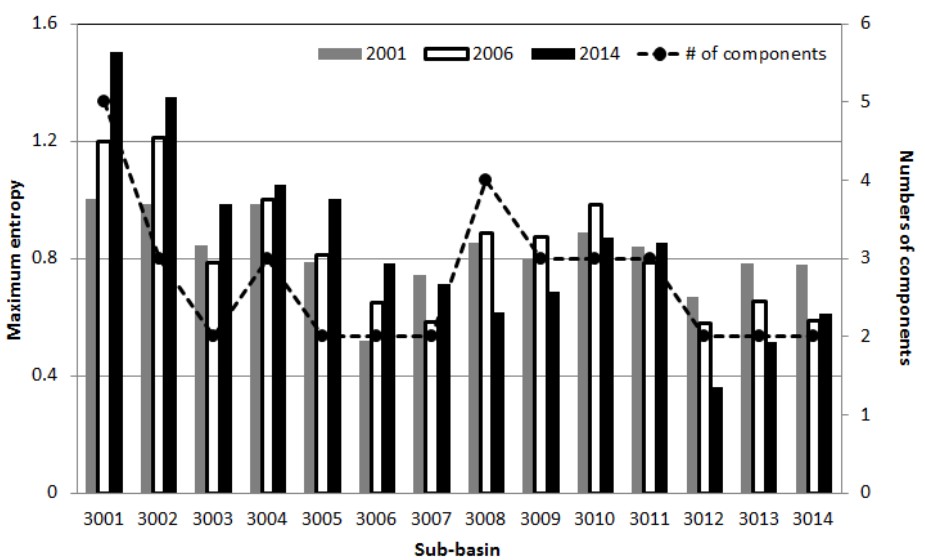

2                                     (a) For each sub-basin

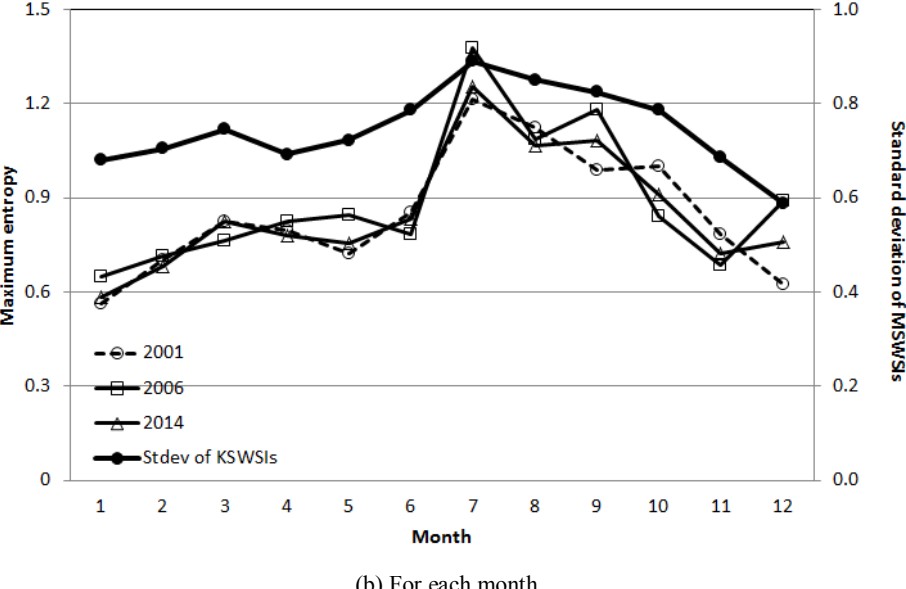

3                                     (b) For each month

5     Fig. 11. Comparison of maximum entropy results between sub-basins and months for each drought

6     event