# Peer review of "The Probabilistic Drought Forecast Based on the Ensemble Technique Using the Korean Surface Water Supply Index"

_Natural Hazards and Earth System Sciences, 2017_

## Referee Comment (RC1) · Anonymous Referee #1 · 14 Dec 2017

For Korea, the authors suggested a new drought index called KSWSI which contains hydrometeorological components (such as Local water supply) which are not considered in the previous drought indices (e.g. MSWSI). Moreover, the probability of hydrometeorological components were estimated from long tail distributions instead of the normal distribution. Although the ideas of this study could be interesting, the current analysis results cannot justify their proposed KSWSI. The authors mainly subjectively claim their KSWSI is better.

Here are three main concerns:

(1) Inadequately setting out the research problem

[Figure]

In the introduction, the authors reviewed previous work. However, they did not motivate their study well. Many arguments are very confusing and subjective. For example, in P3 Lns 5-7, I don't understand why "the dry flood season caused by global warning". I don't know why mitigating droughts, improving drought indices and various conditions are related (P3 Lns 9-11). In the introduction, I would like to know the criteria of good drought indices. It seems that the authors know that intensity and duration are important (P4 Ln 1). However, the authors did not look at them. I am not sure whether the authors looked at "drought forecast" (P4 Ln 9; P2 Ln 11) or just "probability quantification". They know that lead time is important for drought forecast (P4 Ln 15), but they did not look at their forecast performance based on different lead times. Using different literature, the authors should explain why they want to include different new components individually. Based on other works, they should put some efforts to explain why the selection of distributions for different components is important. They need to justify why they only look at the Average Hit Score (AHS) and the Half Brier Score(HBS). They also should try to explain why they would like to look at maximum entropy principles.

(2) Poor analysis

The authors did not verify their results (P2 Ln 8). They just subjectively claim their results. The authors have many subjective and tautological statements. For example, "the accuracy of the drought forecasts using KSWSIs were higher than those using previous SWSI, demonstrating that KSWSI is able to enhance the accuracy of drought forecasts" (P2 Lns 10-12). After reading the whole section 2, I still don't know how they select appropriate variables to be the components KSWSI for different subbasins (P2 Lns 4-6). The authors claim that they investigate "all available hydrometeorological components" (P7 Lns 14-15). It appears to be impossible. I also don't know how they select distributions for their hydrometeorological components. At the moment, the authors assume that the most influential hydrometeorological components (P8 Lns 8-9) for their index would not change over year. The authors need a sensitivity study to demonstrate that. For Figure 8, I may suggest that KSWSI is overestimating drought

risk because its boxplots cannot contain the actual droughts. Based on Figure 8, I would say MSWI is better than KSWSI. The discussion in P14 Lns 11-25 is very problematic. The whole validation of KSWSI is very subjective and qualitive. Here are some examples: In P17 Lns 2-3, the authors suggested that the "drought was avoided due to the abundant water resources". In P17 Lns 17-18, the authors just subjectively guessed that "it is more reasonable that hydrological droughts occurred because of the low precipitation and dam inflow." In P18 Lns 21-26, the authors just subjectively selected their probability distributions. Moreover, I don't understand why a larger standard deviation of KSWSI suggests that the selection of probability distribution is correct (P2 Lns 17-20). Authors' concepts of accuracy seem to mean larger uncertainty (P21 Lns 14-21; P22 Lns 4-7).

(3) Poor organisation

The details of the Average Hit Score (AHS) and the Half Brier Score(HBS) (P13 Lns 17-18) should be in the method section. The details of the maximum entropy principle (P20 Lns 7-24) should be in the method section. The conclusion is somewhat badly written. In P22 Ln 15, the authors just use a vague term, "limitations". The authors need to think how to present their data, arguments and results logically.

---

## Referee Comment (RC2) · Anonymous Referee #2 · 6 Mar 2018

This study proposes a new hydrological index to monitor and forecast drought events in Korea by using an improving version of the Surface Water Supply Index. The authors first compare the behaviour of the indicators for past events and then the forecast scores. Because of a lag in the scientific robustness of the results, of mistakes and wrong methodologies to compare the indices and their forecasts that reduce the scientific significance of the results, and because most of the results are too speculative, I am afraid to reject the paper. For the following version, I also recommend to strongly reduce the size of the document by adding most of the graph and descriptions in supplementary materials.

Here is a non-exhaustive list of major comments:

Introduction: The authors provide a large list of indices used in the literature but not used here. Either the authors add these in this study or they have to simplify the introduction. Also I found the quality of the introduction quite low. The state of the arts and the ongoing questions that motivate this study are missing. I recommend to completely reconsider this section.

Also to justify the fact that their index is more adapted over Korea, the authors should take into account other well known and commonly used indices they describe in the introduction and make a fair and objective comparison.

Figure 1, location of this basin in Korea ? Need to zoom out, orography ?

p4 L9 "Drought forecast ..." Which drought ?

Table 1b not clear at all.

Figure 2 Not visible

Figure 3 put in supp. materials.

Section 2.3 and figure 4 : Is it monitoring or forecasting ? as I understood it is monitoring. Please clarify.

Figure 5 + p10 L1-14. This is too descriptive, also without the climatology of each index, it is useless. What are the distribution and time variability of each value. How the authors 'validate' these graphs. It is really too speculative for me. I did not find scientific significance in these results.

P11 L2 how the authors can conclude that one method is more accurate? Again, all this section is too descriptive and the 'comparison' is done without scientific objectivity. That is why I consider all the conclusions too speculative without scientific evidences.

P11 L11 L14 what are the "scenarios" the authors mention? are there hindcasts?

Please clarify

Table 4 and its description (P12 L8) should be put in Supp. materials.

In section 3.2 why the forecast validation do not include 2001?

All this section of verification is done without scientific objectivity. Do we have to compare Fig 7 with Fig. 4? But if the two indices are different, with different behaviour and climatology, how the authors can compare these forecasts? A better way to assess these forecasts is to compare their improvements related to the climatology of each index (skill scores). Also I found a large part of this section too descriptive and could be improved.

Section 4: I found all this section quite hard to read. The authors propose to assess the uncertainty analysis of the calculation procedure for the KSWSI but, to me, this is not done properly. First the period of study is too short, what is the robustness of these few years? I do not understand why the authors use only few stations. A better way to do this study is to work with a longer time serie and all the stations available to provide robust statistics. Also, I was lost in that section that is too descriptive and too long. I suggest to completely reconsider this section.

According to all these recommendations, the authors should change the conclusions provided.

---

## Author Comment (AC1) · 26 Mar 2018

Authors agreed with the reviewer's comments. The manuscript has been revised as attached file.

Please also note the supplement to this comment:
https://www.nat-hazards-earth-syst-sci-discuss.net/nhess-2017-163/nhess-2017-163-AC1-supplement.pdf

---

## Author Comment (AC2) · 26 Mar 2018

The Probabilistic Drought Forecast Using the Korean Surface Water Supply

                                    Index

Suk Hwan Jang[1], Jae-Kyoung Lee[2], Ji Hwan Oh[3], Jun Won Jo[4], and Younghyun Cho[5]

1. Professor, Department of Civil Engineering, Daejin University, Pocheon-si, Gyeonggi-do,

Korea, drjang@daejin.ac.kr

2. Assistant Professor, Innovation Center for Engineering Education, Daejin University,

Pocheon-si, Gyeonggi-do, Korea, myroom1@daejin.ac.kr

3. Ph.D Candidate, Department of Civil Engineering, Daejin University, Pocheon-si,

Gyeonggi-do, Korea, ojh4525@naver.com

4. Master course, Department of Civil Engineering, Daejin University, Pocheon-si,

Gyeonggi-do, Korea, yhjowon@naver.com

5. Principal Researcher, Hydrometeorological Cooperation Center, Gwacheon-si, Gyeonggi- do, Korea, yhcho@kwater.or.kr

Corresponding Author: Jae-Kyoung Lee, Innovation Center for Engineering Education,

Daejin University, Pocheon-si, Gyeonggi-do, Korea

E-mail : myroom1@daejin.ac.kr

Abstract

Drought due to the shortage of agricultural water damaged throughout the Korean Peninsula in 2014-

2015. In order to effectively mitigate these drought damages, improvement and development of drought indices suitable to Korea should be prioritized to monitor the drought conditions accurately.

This study proposes the new hydrological drought index, Korean Surface Water Supply Index (KSWSI), which overcomes some of the limitations in the calculation procedure of modified SWSI

applied in Korea and conducts the probabilistic drought forecasts using KSWSI. In this study, all hydrometeorological variables in the Geum River basin were investigated and appropriate four to six variables were selected as drought components in KSWSI for each sub-basin. And whereby only the normal distributions are applied to all drought components, probability distributions applicable for each drought component in KSWSI were estimated. As a result of verifying KSWSI results using observed hydrometeorological data, the accuracy of KSWSI showed better drought phenomenon in drought events than MSWSI. The monthly probabilistic drought forecasts were also calculated based on ensemble technique using KSWSI. In 2006 and 2014 drought events, the accuracy of the drought forecasts using KSWSI were higher in both Average Hit Scores (AHS) and Half Brier Scores (HBS)

than those using MSWSI, demonstrating that KSWSI is able to enhance the accuracy of drought forecasts. The influence of expanding hydrometeorological variables and selecting appropriate probability distributions for each drought component of KSWSI were also analyzed. It is confirmed that the accuracy of KSWSI results may be affected by the choice of hydrometerological variables, the station data obtained, the length of used data for each station, and the probability distributions selected. Furthermore, the uncertainty quantification of KSWSI calculation procedure was also carried out using the Maximum Entropy (ME) theory. Estimating appropriate probability distributions for each drought component in the flood season is very crucial because ME values (=1.053 on average)

and standard deviations of KSWSI (=0.843 on average) are very huge, implying that large uncertainty occurs in the flood season.

[revised manuscript text omitted]

Therefore, this study proposes the new and improved hydrological drought index for the accurate monitoring and conducts the methodology to forecast monthly droughts for the Korean Peninsula as follows: Firstly, this study analyzes the limitations of the existing hydrological drought index, Surface Water Supply Index (SWSI), which was applied in the Korean Peninsula and improves and applies the drought index called the Korean Surface Water Supply Index (KSWSI). Secondly, the monthly droughts are forecasted using the improved drought index. The probabilistic monthly drought forecasts are conducted based on the ensemble technique to capture the inherent monthly forecasting uncertainty. Lastly, the effect of the selection of drought components and their probability distributions is analyzed and a method is proposed to quantify their uncertainties.

2. Improvement of hydrological drought index: Korean Surface Water Supply Index

The Surface Water Supply Index (SWSI) (Shafer and Dezman, 1982) was selected as the well- known hydrological drought index. SWSI is advantageous as it can flexibly utilize various hydrometeorological variables depending on the basins. SWSI is based on probability distributions of monthly time series of individual component indices and is calculated using four hydrometeorological variables as drought components: snowpack, precipitation, streamflow, and reservoir storage. It is also an appropriate drought indicator in snow-dominated regions. The drought classification of SWSI is divided into seven categories (extremely dry (-4.2 to -3.0; 7[th] category), moderately dry (-2.9 to -2.0;

6[th] category), slightly dry (-1.9 to -1.0; 5[th] category), near average (-0.9 to 1.0; 4[th] category), slightly wet (1.1 to 2.0; 3[rd] category), moderately wet (2.1 to 3.0; 2[nd] category), and extremely wet (3.1 to 4.2;

1[st] category)) and is similar to the typical categories of the Palmer Drought Severity Index (PDSI).

The mathematical formulation of SWSI is given by:

$$\text{SWSI}_t = \frac{w_1 P_t^{snow} + w_2 P_t^{prec} + w_3 P_t^{strm} + w_4 P_t^{resv} - 50}{12} \tag{1}$$

where $w_1$, $w_2$, $w_3$, and $w_4$ are the weights for each dorught component and $w_1 + w_2 + w_3 + w_4 = 1$, and where $t$ represents the monthly time-step. $P_t^i$ is the non-exceedance probability (in percentage) for component $i$ where the superscripts of *snow*, *prec*, *strm*, and *resv* represent the snowpack, precipitation, streamflow, and reservoir storage in time $t$, respectively. In calculating the SWSI, depending on regions, a snowpack component is applied from December to the subsequent May, and a streamflow component is applied during the remaining periods. Kwon et al. (2006) and Kwon and

Kim (2006) then developed a Modified SWSI (called MSWSI) by improving SWSI for the Korean

Peninsula. In MSWSI, the snowpack component is replaced by groundwater because the portion of underground water is more important to snowpack in the water resources management in Korea:

$$\text{MSWSI}_t = \frac{w_1 P_t^{gw} + w_2 P_t^{prec} + w_3 P_t^{strm} + w_4 P_t^{resv} - 50}{12} \qquad (2)$$

where *gw* represents the groundwater component. The process of MSWSI calculation is as follows:

Step 1: Analysis of available hydrometeorological variables by basins

Step 2: Selection of available hydrometeorological variables as drought components and collection of observed data

Step 3: Calculation of weights for each drought component

Step 4: Estimation of probability distributions for each drought component

Step 5: Calculation of MSWSI values using Eq. (2)

However, this process of MSWSI calculation has several limitations. Firstly, only four hydrometeorological variables are used in the previous MSWSI calculation in Steps 1 & 2 and the

MSWSI is not able to reflect more various variables. Different hydrometeorological variables actually impact drought events depending on data length, the urban area, and upstream & downstream areas of dams; therefore, the available variables should be widely investigated. Secondly, in Step 4, probability distributions of all hydrometeorological variables were fitted to the only normal distribution in the

MSWSI calculation process. Estimating the appropriate probability distribution for each variable yields accurate non-exceedance probability values, which can be used to estimate the near actual drought index.

Therefore, in this study, an improved MSWSI was developed, called the Korean SWSI (KSWSI), with two improvements. The first improvement involves investigating all available hydrometeorological variables for each sub-basin and selecting the appropriate variables as drought components. The second improvement involves estimating and applying a suitable probability distribution for each selected hydrometeorological variable. The detailed improvements are as described in the following section and Fig. 1 shows the process of the MSWSI calculation and its improvements.

[Fig. 1. Procedure of KSWSI calculation and two improvements proposed by this study]

2.1 Study basin

This section describes the Geum River basin as the applicable area for improving the drought index and verifying the drought forecast (Fig. 2). The Geum River basin flows north-westerly to about its mid-point, then generally south-westerly for 401km. It consists of 21 sub-basins, and drains into an area of 9,810km$^2$. The Geum River basin has two multi-purpose dams, Daecheong Dam and

Yongdam Dam. Daecheong Dam provides municipal and industrial water supply to Daejeon and

Chungju, and Yongdam Dam (which is only one-fifth the size of the Daecheong Dam drainage area)

supplies water to Jeonju. Analyzing the river flow in the Geum River basin is relatively simple because it has fewer dams and a simpler river system than other basins. The region of the Geum River basin has been affected by considerable drought since 2000 year and has been widely used in previous drought studies in Korea.

            [Fig. 2. Study basin: 14 sub-basins in Geum River basin]

2.2 Selection of available hydrometeorological variables as drought components

In previous drought studies in Korea, as mentioned, MSWSI results were calculated using only four hydrometeorological variables. MSWSI cannot demonstrate the actual drought accurately because of limitations of practical data. . The values of the previous MSWSI are also calculated using a finite number of observation stations: precipitation data obtained from six stations, streamflow data obtained from 10 stations, groundwater data obtained from 3 stations, and only dam inflow data for only one dam.

In this study, all hydrometeorological data from each sub-basin in the Geum River basin were investigated and classified into 9 types: precipitation data, water level data in dam, meteorological data, national streamflow data, local streamflow levels 1 & 2 data, multi-regional water supply, local water supply, and groundwater (Table 1(a)). The precipitation data, water level data, water discharge data, streamflow data, dam data (included in inflow, release, and storage data), and groundwater data were selected as practical hydrometeorological variables on the basis of ease of data acquisition, data quality control, and data length. These data were then collected from (areal-averaged) precipitation data from 42 stations, streamflow data from 28 stations, groundwater data from 7 stations, and dam data included in inflow, release, and storage data (Table 1(b)).

Table 2 shows the final hydrometeorological variables and stations selected as drought components for each sub-basin. The sub-basins were also classified into dam inflow, dam water-level, streamflow, groundwater, precipitation, and water supply-dominant basin depending on the most influential drought component that has the largest monthly-averaged weight for each sub-basin. Doesken et al. (1991) proposed a method that can reflect the relative contribution of drought components to estimate the weights ($w_1$, $w_2$, $w_3$, and $w_4$). The initial weights of each month for each component were calculated as monthly values divided by the annual total of the component. The calculated monthly values of selected components of KSWSI were summed for each month. Then, the twelve monthly sums, calculated using this procedure, were divided by their total sum to find the sum of the final weights as 1. As shown in Fig. 3, a dam component has an important impact, relatively, in sub-basin 3001 located in the upstream of Yongdam dam and sub-basin 3007 were affected by precipitation and streamflow because of similar averaged weights. Especially, the effects of streamflow and precipitation components are varied slightly month by month, with the effect of the precipitation component being greater in the flood season overall.

[Table 1. Basic investigation of hydrometerological variables for each sub-basin]

[Table 2. Selected hydrometerological variables and stations for each sub-basin]

[Fig. 3. Example of weights of each drought component for each month at sub-basin 3001 and

3007]

2.3 Estimation of suitable probability distribution for each drought component

Drought studies using MSWSI fitted all drought components to the only normal distribution.

These MSWSI results could not accurately simulate the actual droughts. In this study, the probability distributions (Generalized Extreme Value (GEV), Gumbel, normal, 2-parameter log-normal, log- normal, and 3-parameter log-normal distribution) applicable to each drought component and parameter estimation methods (e.g. maximum likelihood method, probability weighted moment method, and method of moment) are applied and then log-likelihood test is also used for the goodness of fit test. Table 3 shows final selected probability distributions for drought component for each sub- basin.

[Table 3. Selected suitable probability distributions to drought components for each sub-basin]

2.4 Application of KSWSI

In this study, 2001, 2006, and 2014-year events were used, when the severe drought occurred nationally. In the 2001 event, the average rainfall amounts were as high as 377mm from March to

May, which was 20%~30% of the annual rainfall amounts in some regions in Korea. The rainfall amounts from August to October was only 30% of the annual rainfall amounts in the south part of the

Korean Peninsula in 2006 and the national reservoir storage ratio was 67% on average (NEMA, 2009).

In 2014, a severe drought occurred in northern Korea, where average rainfall amounts were 50%~61%

compared to the normal-year, where the normal-year is the mean of the last 30-year average rainfall (KMA, 2014).

*Comparison of MSWSIs and KSWSIs in sub-basin 3001*

The verification of drought indices is practically restricted. In this study, the accuracy of KSWSI

was indirectly determined using the tendency of observed hydrometeorological variables. Fig. 4

shows the results of the MSWSI and KSWSI for April in 2001, 2006, and 2014 in Geum River basin.

In 2001, both MSWSI and KSWSI generally showed a similar drought trend; while the MSWSI in the

Daecheong Dam had moderate and extreme droughts, the KSWSIs showed near normal and slight droughts. In 2006 and 2014, the KSWSIs showed stronger drought intensities in some sub-basins than the MSWSIs; especially, KSWSIs indicated that droughts in the western sub-basins were more severe.

Fig. 5(a) shows the time series for the MSWSIs and the KSWSIs in sub-basin 3001 for the 2014

event. In the MSWSI, slightly severe or severe droughts were simulated to occur continuously; however, KSWSIs were overall above the near normal droughts. Fig. 5(b) shows the time-series of non-dimensional ratios to the normal droughts during in the 2013-2014 years for each hydrometeorological variable such as precipitation, streamflow, and dam inflow. In block A of Fig.

5(a) and block A1 of Fig. 5(b), the ratios of precipitation and dam inflow were lower than the normal- year in January-February 2014, but inflows and streamflow were abundant due to the increased precipitation (up to 164%) compared to the normal-year from September to December 2013. As these effects continued until early 2014, it is more reasonable to assume that hydrological drought did not occur in sub-basin 3001. In the flood season, the amount of precipitation and dam inflow were lower than the normal-year, but water shortage did not occur due to the abundant precipitations from March to April. In block B of Fig. 5(a) and block B1 of Fig. 5(b), MSWSI showed sub-basin 3001 under drought conditions, but the dam inflow and streamflow increased due to the significantly higher precipitation than normal-year, and KSWSIs showed that sub-basin 3001 was more moderately wet.

*Comparison of MSWSIs and KSWSIs in sub-basin 3014*

Fig. 5(c) shows the time series for the MSWSIs and the KSWSIs in sub-basin 3014 for the 2001

event. The MSWSIs were somewhat varied; however, most of them were above the normal drought level and no dry condition occurred, except in July and August. On the other hand, in the KSWSIs, most of the droughts occurred in 2001, and severe drought occurred in early 2001. Fig. 5(d) shows the time-series of the non-dimensional ratios to the normal-year during the 2001-2002 years for each hydrometeorological component such as precipitation, streamflow, and dam inflow. In block C in Fig.

5(c) and block C1 in Fig. 5(d), the amount of observed precipitation and streamflow, which were only

40%~60% of the normal-year, contributed to the water storage, resulting in severe drought. Therefore, it is more reasonable to conclude that hydrological drought occurred in sub-basin 3014.

As shown in the previous examples, compared to the MSWSIs, the KSWSIs calculated more accurate drought results in the Geum River basin. Therefore, it is confirmed that the KSWSI is more appropriate in hydrological drought monitoring and forecasting.

[Fig. 4. Comparison of MSWSI and KSWSI results in April 2001, 2006, and 2014 drought events]

[Fig. 5. Verification of KSWSI results in sub-basin 3001 and 3014 in 2001 and 2014 drought events]

3. Monthly Probabilistic Drought Forecasts

3.1 Application outline

This study considered 16 historical scenarios (1990~2005) and 24 historical scenarios (1990~2013) with variables of drought components for monthly drought forecast for 2006 and 2014, respectively. For drought forecasting to January 2006, for example, 16 historical scenarios (1990~2000) of precipitation and temperature were inputted into hydrological models to generate streamflows and groundwater level ensembles. For each forecasting period, the hydrological model was executed with the hydrometeorological variables for the preceding 12 months to determine the initial conditions. The historical data of each drought component were then fitted to their proper probability distribution to make the variable dimensionless. These ensembles finally served as inputs in the calculation of the values of KSWSI with their weights. Fig. 6 shows the procedure of monthly probabilistic drought forecasts.

In this study, the accuracy of the probabilistic forecast was measured using the Average Hit Score (AHS) and Half Brier Score (HBS) (Wilks, 1995). The AHS scored the probabilities of occurrences of drought forecasts for the drought category by the actual drought, and the ensemble drought forecasts can be considered to be effective if their AHS is higher than the AHS of the naive forecasts. The concept of HBS is similar to the mean square error and is a way to give a high score when ensemble drought forecasts match the actual drought, but gives a penalty for wrong categories. The drought forecast becomes increasingly more accurate as the HBS becomes smaller than the naive forecast. The equations of AHS and HBS are as follows:

$\text{AHS} = \dfrac{1}{N} \sum\limits_{i=1}^{N} f_i^o$                       (3)

$\text{HBS} = \dfrac{1}{N} \sum\limits_{j=1}^{J} \sum\limits_{i=1}^{N} (f_{i,j} - o_{i,j})^2$              (4)

where $f^o$ is the probability of drought forecast for the category of actual drought, $N$ is the number of drought forecasts, $J$ is the number of drought categories, $f_{i,j}$ is the probability of the $i$th forecast in the

$j$th category, and $o_{i,j}$ is the actual drought in the $j$th category. The category of actual drought score is 1

at the $i$th drought forecast and the scores of the remaining categories are zero.

*Calibration of the hydrological model*

In this study, the *abcd* water balance model was used, which has parameters of *a*, *b*, *c*, and *d* to determine the streamflow and groundwater. The parameters of the *abcd* model are estimated with a regional regression for ungauged basins because streamflow is gauged only at Yongdam and

Daecheng Dams. The regional regression equation was then formulated using the relationship between each of the calibrated parameters and the site specific basin characteristics such as basin length, drainage area, basin annual average precipitation, basin annual average potential evapotranspiration, basin average land height, basin average land slope, basin drainage density, basin average temperature, basin monthly maximum precipitation, basin monthly maximum potential evapotranspiration, drainage relief, soil type, and basin total stream length. The calibrated parameters,

*a*, *b*, *c*, and *d* of the *abcd* model were obtained using gauged stations in nine multipurpose dams in

Korea. Table 4 shows the regional regression equations over all of Korea as a result of a step-wise regression technique. Using these equations with basin characteristics of an ungauged basin, *a*, *b*, *c*, and *d* can be computed and consequently the streamflow of the basin can be computed from the calibrated *abcd* model.

To verify the estimated parameters of the *abcd* model using the regional regression equations, the

*abcd* model was applied to generate the monthly inflows at Yongdam Dam from 2002 to 2004 (period

#1) and from 2010 to 2013 (period #2). The calculated values of the R-Bias, R-RMSE, and $R^2$ during period #1 were -0.06, 35, and 0.92, respectively, and those during period #2 were 0.11, 0.55, and 0.91, respectively, suggesting that the model parameters are accurately estimated.

[Fig. 6. Example of the procedure of the monthly probabilistic drought forecast]

 [Table 4. Regression equations for the *a*, *b*, *c*, and *d* parameters]

3.2 Results of monthly drought forecasts

Fig. 7 showed the monthly drought forecasts using MSWSI and KSWSI in April and December of 2006 and 2014, respectively. Drought-intensities in the drought forecasts using the KSWSI were stronger than in the MSWSI, and the drought occurred widely throughout the Geum River basin.

While the MSWSI-based drought forecasts for April 2006 and 2014 predicted slight and moderate drought in some sub-basins of downstream and near Yongdam Dam, the results of the KSWSI

forecasted severe and moderate droughts in most sub-basins of the Geum River basin. Then, in

December 2006 and 2014, drought forecasts of MSWSI were similar to those of KSWSI; especially, in December 2014, drought forecasts by KSWSI showed severe droughts in some sub-basins of downstream and near Yongdam Dam. Table 5 shows the occurrence probabilities of droughts for each sub-basin by drought forecast using MSWSI and KSWSI for April and December 2014. From the drought forecasts using KSWSI, the probabilities of severe droughts in both April and December 2014 were over 70%, showing droughts were highly likely to occur.

The drought forecasts were compared to the corresponding observed event for a verification period of 12 months in 2006 and 2014. As shown in Table 6(a), the AHS of the 2006 and 2014 events are 0.201 and 0.200, respectively, which are higher than that of the naive forecast (=0.174). Especially, the AHSs of drought forecasts using KSWSI are 0.249 and 0.325 for 2006 and 2014, respectively, which is more accurate than the drought forecast using MSWSI. The overall accuracy of the drought forecasts was better in the dry season (October to the following May) than in the flood season (from July to September), and the accuracy of drought forecasts using KSWSI was improved from 0.219 to 0.397 by AHS. As shown in Table 6(b), while the accuracy of drought forecasts using MSWSI is 0.848 in 2006, which is smaller than that of the naive forecast (=0.857) for 2006 and 2014, the accuracy of MSWSI in 2014 (=0.865) was low. The accuracy of drought forecasts using KSWSI was confirmed to be superior to that of the MSWSI because HBSs of KSWSI are 0.824 and 0.795 in 2006 and 2014, respectively. The actual drought and occurrence ranges of drought forecasts using MSWSI and KSWSI were compared. Fig. 8(a) shows the monthly actual droughts (black dots) and occurrence ranges of drought forecast ensembles (between the first and third quartiles of the box-plot) from January to December 2014 at sub-basin 3001. The actual droughts exist in the range of the drought forecast ensembles, implying that the drought forecasts consider the extent of the actual drought and as the range of drought forecast ensembles narrows, including the occurrence of actual drought, the accuracy of drought forecasts increases. While the ranges of drought forecasts using MSWSI include several actual droughts, the actual droughts are out of ranges of drought forecasts using KSWSI. As shown in Figs. 8(b) at sub-basin 3007, the drought forecasts with MSWSI are effective because most categories (block A: extremely drought) of drought forecast ensembles include actual droughts. Especially, the right-side box-plots have narrow ranges in the drought forecasts, implying that the ensemble ranges of KSWSI drought forecasts are very concentrated in the category of 'extremely dry'

and the actual droughts also occur in the same category, so that the accuracy of the drought forecasts using KSWSI is superior to MSWSI. Fig. 8(c) at sub-basin 3014 shows a similar tendency to that of sub-basin 3007, confirming the high accuracy of the drought forecast using KSWSI. While actual droughts were more severe than the drought forecasts using MSWSI, KSWSI drought forecasts demonstrate the 'extremely dry', including most of actual droughts.

[Fig. 7. Comparison of the drought forecasts using MSWSI and KSWSI on April and December in

2006 and 2014 drought events]

[Table 5. Comparison of the most probable drought categories and their correspoding probabilities for each sub-basin in April and December on 2014 drought event]

[Table 6. The accuracy of MSWSI and KSWSI forecasts]

[Fig. 8. Comparison of drought forecast ranges for each month at sub-basin 3001, 3007, and 3014 in

2014 drought event]

4. Uncertainty Analysis

4.1 Maximum entropy principle

Shannon (1948) first introduced the use of entropy as a method to estimate uncertainty quantitatively if the information context is obtained from probability distributions of a given set of information. If probabilities of occurrences of a certain set of information are large, the amount of information is small, and if their probabilities are small, the amount of information becomes large. If

$X$ is defined as a random variable with probability $p$, and $I(X)$ is the information context of $X$, entropy

$H(X)$ is given as follows:

$$H(X) = -\sum p_X(x)\ln p_X(x) = \sum p_X(x)I(X) = E[I(X)] \tag{5}$$

Maximum Entropy (ME) based on Shannon's entropy theory (1948) was proposed by Jaynes (1957). When a certain set of information is given, based on the information, maximum entropy theory provides the probability density function which maximizes the entropy. If a given set of information is the minimum value $a$ and maximum value $b$, the distribution maximizing the entropy is a uniform distribution on $[a, b]$, and the corresponding entropy $H(X)$ (i.e. maximum entropy) is given as (Gay and Estrada, 2010):

$$H(X) = -\int_a^b f_X(x) \ln f_X(x) dx = -\int_a^b \frac{1}{b-a} \ln \frac{1}{b-a} dx = -\ln(b-a) \qquad (6)$$

4.2 Occurrence of uncertainty

In steps 1, 2, and 4 in KSWSI calculation process described in Chapter 2, the researcher's experience and subjective judgment are involved. For example, the researchers can select several hydrometeorological variables as drought components and fit the probability distributions to the selected drought components. This means that each researcher has a different choice of variables and distributions because of different experience and criteria. Therefore the final KSWSIs can differ according to the researcher's subjective judgment; this likely results in uncertainty about the drought monitoring and forecasts. The subjective judgments of the researchers for each stage of KSWSI

calculation are as follows.

• *Step 1&2: Analysis and selection of hydrometeorological variables for each basin*

(a) selection of available hydrometeorological variables as drought components (b) data quality verification of selected drought components (c) selection of observation stations to acquire hydrometeorological data as drought components

As mentioned above, in this study, the precipitation data, water level data, discharge data, streamflow data, dam data (included in inflow, release, and storage data), and groundwater data were selected as hydrometeorological components that can be practically applied as KSWSI drought components. Table 7 shows that, for MSWSI, observed data in only one station was used for each drought component (K-water, 2005); however, averaged data were used from several stations in

KSWSI calculation. Especially, in the case of precipitation, areal-averaged data using the Thiessen method was used rather than point data. Secondly, only the data of Daecheong Dam was reflected in

MSWSI, because the data length of Yongdam Dam was insufficient at the time of drought researches using MSWSI. This study used the observation data of dams as follows: (1) for applying dam data, the sub-basins in Geum River basin were divided into those that were affected by Yongdam Dam and those affected by Daecheong Dam; (2) sub-basins around dams were also divided into upstream and downstream sub-basins, and the observation data of dam inflow and storage in the upstream and dam release in downstream were then applied to KSWSI calculation, respectively. Finally, while MSWSI

calculation only reflected four drought components, KSWSI reflected a maximum of six drought components and the number of observation stations used to obtain meteorological data in all drought components was increased.

• *Step 4: Estimation of probability distributions for each drought component*

(a) estimation of probability distributions for each drought component (b) selection of proper probability distributions for each drought component

In Chapter 2, the precipitation component was fitted to the Gumbel and GEV distributions, the normal and Gumbel distributions for streamflow, 2-parameter log-normal and Gumbel distributions for dam data (inflow, release, and storage), and the 3-parameter log-normal distribution for groundwater. Since the drought components which are applied for each sub-basin differ and several probability distributions can be applied in the even same sub-basin, KSWSI results can differ depending on the probability distributions selected. In this study, we determined how the results could be changed by calculating KSWSIs by applying all the probability distributions (including the normal distribution) that are shown to be appropriate.

[Table 7. Comparison of hydrometeorological variables for each sub-basin in drought researches using MSWSI and KSWSI]

4.3 Application

In this section, the influence of researcher's subjective judgment on KSWSI calculation and its corresponding uncertainty are analyzed.

*Analysis of the influence of expanding hydrometeorological components as drought components*

In order to investigate the influence of the selection of hydrometeorological variables, KSWSI results for 2001 and 2006 drought events were calculated using the drought components selected in Table 2. Similar to drought researches using MSWSI (K-water, 2005), the probability distributions of all drought components were assumed to be normal distributions. In Table 8, the results of both MSWSI and KSWSI showed drought as a whole in all of the sub-basins. Especially, the identical MSWSI results were calculated from the same drought components from sub-basin 3001 to 3004, whereas KSWSI results showed slightly different drought values and categories. In the 2006 drought event, while MSWSI indicated that the water resources of the entire Geum River system were very low, resulting in drought. KSWSI demonstrated the contrary results, where drought was avoided due to the abundant water resources.

(1) Comparison of MSWSIs and KSWSIs in sub-basin 3001

Fig. 9(a) shows the time series for MSWSI and KSWSI in sub-basin 3001 for the 2006 drought event. In both MSWSI and KSWSI, drought occurred in the beginning of 2006, whereas the drought was somewhat resolved as the flood season passed. However, the drought-intensity calculated by

KSWSI is stronger than that by MSWSI. Fig. 9(b) shows the time-series of non-dimensional ratios to the normal-year for the 2005-2006 years for precipitation, streamflow, and dam inflow. In block A of

Fig. 9(a) and block A1 of Fig. 9(b), the amount of precipitation and dam inflows were lower than the normal-year from January to April 2005, and streamflow was almost the same as normal-year. In block B of Fig 9(a) and block B1 of Fig. 9(b) in July 2006, the dam inflow and streamflow both increased due to very large precipitation compared to the normal-year, and since August, the dam inflow also decreased because precipitation was very low. For the observed hydrometeorological data for March, June, and August 2006, while the amount of streamflow is maintained, it is more reasonable that hydrological droughts occurred because of the low precipitation and dam inflow.

(2) Comparison of MSWSIs and KSWSIs in sub-basin 3010

Fig. 9(c) shows the time series for MSWSI and KSWSI in sub-basin 3010 for the 2006 drought event. While MSWSI results show no drought in early 2006 except severe droughts in the flood season, KSWSI results are included in below the category of 'near normal', except for July, and indicated that water shortage occurred for the entire period. In block C of Fig 9(c) and block C1 of Fig.

9(d), MSWSI results indicated that water resources were abundant, but some water shortages did actually occur, and the accuracy of KSWSI results is considered to be superior to that of MSWSI

because precipitation is very influential in this season. In block D of Fig. 9(c) and block D1 of Fig.

9(d), in July 2006, a large amount of precipitation occurred compared to the normal-year, so the amount of both dam release and streamflow was increased and the water shortage was then resolved.

After August, the amounts of both dam release and streamflow decreased. MSWSI results showed severe drought in July when the amount of precipitation, streamflow, and dam release were larger than normal-year, but KSWSI results indicated that the drought was resolved. In 2006, the streamflow and dam release were smaller than normal-year and their variation was not significant. Reflecting the water resources, KSWSI showed that droughts were resolved due to the occurrence of precipitation, but water shortages had generally occurred.

As shown in the previous results, the KSWSI may affect whether or not the actual droughts are accurately simulated by KSWSI calculation depending on the hydrometerological variables as the drought components, which station data are obtained, and the length of used data for each station, respectively.

*Analysis of the influence of the probability distribution selection for each drought component*

Table 9 shows applicable probability distributions to each drought component. In the application process, the maximum number of scenarios for probability distributions applicable to each sub-basin is 36 (= 3 probability distributions for precipitation × 2 for river flow × 3 for dam data × 2 for groundwater), and the ranges of KSWSI results are indicated using the maximum and minimum values among these combinations in Fig. 10.

Fig. 10(a) represents the maximum and minimum time series of KSWSI results showed a similar tendency in the 2006 drought event, but the maximum series of KSWSI kept the distance by two to three categories from the minimum. The maximum time series of KSWSI was located above the category 'near normal', which means droughts did not occur, whereas the minimum values of

KSWSIs showed droughts due to water shortage except for July. The KSWSI using only normal distribution are similar to the averages of the maximum and minimum time series of KSWSI. In the

2014 drought event shown Fig. 10(b), the maximum of KSWSI are also above the category 'near normal', which means the water resources are abundant in 2014; however, the minimum of KSWSI

shows continuous severe drought, similar to time series of KSWSI using only the normal distribution.

In sub-basin 3008 shown Fig. 10(c), the maximum and minimum time series of KSWSI showed similar trends in the 2006 drought event, and the maximum time series of KSWSI had a distance by 2

to 4 categories to the minimum. Furthermore, the maximum of KSWSI did not show water shortages, and the minimum showed droughts in March, August, and September. In Fig. 10(d) for the 2014

drought event, while the maximum time series of KSWSI was almost similar to the minimum from

January to May, the maximum and minimum of KSWSI significantly kept a difference in the flood season.

The scenario ranges of KSWSI generally varied according to the selection of probability distributions, and their results of droughts significantly differed depending on the probability distributions selected for each drought component. Therefore, it was confirmed that the selection of the probability distributions could affect the accuracy of results of the KSWSI calculation.

[Table 8. Comparison of MSWSI and KSWSI results in July for each sub-basin]

[Fig. 9. Verification of MSWSI and KSWSI results in sub-basins 3001 and 3010: (a) & (b) at 3001

and (c) & (d) at 3010]

[Table 9. Applicable probability distributions for each drought component at each sub-basin]

[Fig. 10. Comparison of the maximum and minimum time series of KSWSI at sub-basin 3001 and

2008 in 2006 and 2014 drought events: (a) & (b) at 3001 & 3008, respectively, in 2006 and (c) & (d)

at 3001 & 3008, respectively, in 2014]

4.4 Quantification and analysis of uncertainty

In this section, KSWSI results calculated by selected drought components and their corresponding probability distributions in Section 4.3 are inputted into the formula (Eq. (6)) of ME to estimate and analyze uncertainty of KSWSI results shown in Table 10 and Fig. 11. Of the ME values for each sub-basin in Table 10(a), the ME value (=1.002) in sub-basin 3001 is the largest and the minimum ME is 0.521 in sub-basin 3006 in the 2001 drought event. In 2006 and 2014 drought event, at sub-basins 3002 and 3001, uncertainty has a large scale, ME values of 1.120 and 1.503, respectively, whereas the smallest ME values of 0.578 and 0.363, respectively, at sub-basin 3012.

Especially, even though the ME values of each sub-basin slightly differ, ME values showed a similar tendency in the same sub-basin despite different drought events. This tendency is more evident in the comparison of the number of ME values for each drought event, drought component, and number of selected drought components for each sub-basin. In other words, the ME values of the sub-basins with many drought components are large, and sub-basins with few drought components, have relatively small ME values. The different drought components for each sub-basin include the data of dam inflow, dam release, groundwater, and data of precipitation and streamflow components, and were used in all sub-basins. Because the data of different observation stations was used for each sub-basin, it could not be determined whether the difference of ME values for each sub-basin was more influenced by dam and groundwater components than by precipitation and streamflow. From the above results, it can be deduced that the increased number of drought components does not necessarily improve the accuracy of the KSWSIs calculation to the actual droughts. In other words, the large values of MEs imply that the results of KSWS have large uncertainty. Therefore, only drought components that can represent the hydrometeorological characteristic of each sub-basin should be selected and applied.

In the monthly MEs for each drought event in Table 10(b), the ME values (1.215 and 1.379) in July are the maximum and the minimum ME at 0.562 and 0.650 in January in the 2001 and 2006 drought events, respectively. In 2014 drought event, the seasonal ME value was the largest at 1.053 in the flood season. Furthermore, in all drought events, although ME values decreased in the dry season, they increased in the flood season as shown in Fig 11(b). To determine the reasons for this result, the standard deviations of KSWSI results according to the selected probability distributions in Section 2.4

are also shown in Fig. 11(b). The trend of standard deviations of KSWSI results was similar to the monthly ME values for each drought event, which decreased in the dry season and increased in the flood season. The large standard deviations of KSWSI results mean that the variation of calculated

KSWSI results depending on the selection of probability distributions is large, which affects the uncertainty of KSWSI results. In other words, applying the appropriate probability distributions to selected drought components in the flood season is very crucial because ME values and standard deviations of KSWSI are very large, implying that huge uncertainty occurs in the flood season.

[revised manuscript text omitted]

| Basin No. | KSWSI components | | | |
|---|---|---|---|---|
| | Precipitation | Streamflow | (related to) Dam | Groundwater |
| 3001 | ·Gumbel
·GEV
·Normal | ·Gumbel
·Normal | ·2-Log-Normal
·Gumbel
·Normal | ·3-Log-Normal
·Normal |
| 3002 | ·Gumbel
·GEV
·Normal | ·Gumbel
·Normal | ·2-Log-Normal
·Gumbel
·Normal | |
| 3003 | ·Gumbel
·GEV
·Normal | ·Gumbel
·Normal | | |
| 3004 | ·Gumbel
·GEV
·Normal | ·Gumbel
·Normal | | ·3-Log-Normal
·Normal |
| 3005 | ·Gumbel
·GEV
·Normal | ·Gumbel
·Normal | | |
| 3006 | ·Gumbel
·GEV
·Normal | ·Gumbel
·Normal | | |
| 3007 | ·Gumbel
·GEV
·Normal | ·Gumbel
·Normal | | |
| 3008 | ·Gumbel
·GEV
·Normal | ·Gumbel
·Normal | ·2-Log-Normal
·Gumbel
·Normal | |
| 3009 | ·Gumbel
·GEV
·Normal | ·Gumbel
·Normal | | ·3-Log-Normal
·Normal |
| 3010 | ·Gumbel
·GEV
·Normal | ·Gumbel
·Normal | ·2-Log-Normal
·Gumbel
·Normal | |
| 3011 | ·Gumbel
·GEV
·Normal | ·Gumbel
·Normal | | ·3-Log-Normal
·Normal |
| 3012 | ·Gumbel
·GEV
·Normal | ·Gumbel
·Normal | | |
| 3013 | ·Gumbel
·GEV
·Normal | ·Gumbel
·Normal | | |
| 3014 | ·Gumbel
·GEV
·Normal | ·Gumbel
·Normal | | |

Table 10. Maximum entropy results for each sub-basin and month in each drought event (a) For each sub-basin

| Basin No. | Maximum entropy | | | Average |
|---|---|---|---|---|
| | 2001 | 2006 | 2014 | |
| 3001 | 1.002 | 1.198 | 1.503 | 1.234 |
| 3002 | 0.985 | 1.210 | 1.352 | 1.182 |
| 3003 | 0.845 | 0.785 | 0.985 | 0.872 |
| 3004 | 0.985 | 1.002 | 1.052 | 1.013 |
| 3005 | 0.789 | 0.812 | 1.005 | 0.869 |
| 3006 | 0.521 | 0.651 | 0.785 | 0.652 |
| 3007 | 0.742 | 0.584 | 0.712 | 0.679 |
| 3008 | 0.854 | 0.888 | 0.616 | 0.786 |
| 3009 | 0.795 | 0.875 | 0.687 | 0.786 |
| 3010 | 0.891 | 0.985 | 0.871 | 0.916 |
| 3011 | 0.841 | 0.784 | 0.852 | 0.826 |
| 3012 | 0.668 | 0.578 | 0.363 | 0.537 |
| 3013 | 0.784 | 0.652 | 0.514 | 0.650 |
| 3014 | 0.781 | 0.587 | 0.612 | 0.660 |

(b) For each month

| Month | Maximum entropy | | | Average | Season | Averaged ME |
|---|---|---|---|---|---|---|
| | 2001 | 2006 | 2014 | | | |
| 1 | 0.562 | 0.650 | 0.541 | 0.584 | Spring | 0.787 |
| 2 | 0.701 | 0.716 | 0.629 | 0.682 | | |
| 3 | 0.825 | 0.765 | 0.882 | 0.824 | | |
| 4 | 0.795 | 0.827 | 0.722 | 0.781 | Summer | 1.053 |
| 5 | 0.721 | 0.847 | 0.697 | 0.755 | | |
| 6 | 0.854 | 0.785 | 0.865 | 0.835 | | |
| 7 | 1.215 | 1.379 | 1.174 | 1.256 | Autumn | 0.904 |
| 8 | 1.125 | 1.087 | 0.992 | 1.068 | | |
| 9 | 0.987 | 1.182 | 1.077 | 1.082 | | |
| 10 | 1.002 | 0.843 | 0.883 | 0.909 | Winter | 0.676 |
| 11 | 0.785 | 0.686 | 0.695 | 0.722 | | |
| 12 | 0.625 | 0.889 | 0.768 | 0.761 | | |

[Figure]

Fig. 1. Procedure of KSWSI calculation and two improvements proposed by this study

[Figure]

Fig. 2. Study basin: 14 sub-basins in Geum River basin

[Figure]

                    (a) Sub-basin 3001

[Figure]

                    (b) Sub-basin 3007
Fig. 3. Example of weights of each drought component for each month at sub-basin 3001 and 3007

[Figure]

April 2001          April 2006          April 2014

(a) MSWSI results

April 2001          April 2006          April 2014

(b) KSWSI results

Fig. 4. Comparison of MSWSI and KSWSI results in April 2001, 2006, and 2014 drought events

[Figure]

(a) Monthly time series of MSWSI and KSWSI at sub-basin 3001 in 2014 drought event

(b) Monthly time series of precipitation, water level, and dam inflow at sub-basin 3001 in 2013-2014

(c) Monthly time series of MSWSI and KSWSI at sub-basin 3014 in 2001 drought event

[Figure]

(d) Monthly time series of monthly precipitation and streamflow at sub-basin 3014 in 2000-2001

Fig. 5. Verification of KSWSI in sub-basin 3001 and 3014 in 2001 and 2014 drought events: (a) & (b)

at 3001 and (c) & (d) at 3014

[Figure]

Fig. 6. Example of the procedure of the monthly probabilistic drought forecast

| Date | MSWSI | KSWSI |
|------|-------|-------|

[Figure]

Fig. 7. Comparison of the drought forecasts using MSWSI and KSWSI on April and December in

2006 and 2014 drought events

(a) MSWSI (left) and KSWSI (right) forecasts at 3001

(b) MSWSI (left) and KSWSI (right) forecasts 3007

(c) MSWSI (left) and KSWSI (right) forecasts at 3014

Fig. 8. Comparison of drought forecasts ranges for each month at sub-basin 3001, 3007, and 3014 in

2014 drought events

(a) Monthly time series of MSWSI and KSWSI at sub-basin 3001 in 2006 drought event

(b) Monthly time series of precipitation, streamflow, and dam inflow at sub-basin 3001 in 2005-2006

(c) Monthly time series of MSWSI and KSWSI at sub-basin 3010 in 2006 drought event

[Figure]

(d) Monthly time series of precipitation, streamflow, and dam release at sub-basin 3010 in 2005-2006

Fig. 9. Verification of MSWSI and KSWSI results in sub-basins 3001 and 3010 in 2006 drought event:

(a) & (b) at 3001 and (c) & (d) at 3010

(a) Sub-basin 3001 in 2006         (b) Sub-basin 3008 in 2006

(c) Sub-basin 3001 in 2014         (d) Sub-basin 3008 in 2014

Fig. 10. Comparison of the maximum and minimum time series of KSWSI at sub-basin 3001 and

3008 in 2006 and 2014 drought events: (a) & (b) at 3001 & 3008, respectively, in 2006 and (c) & (d)

at 3001 & 3008, respectively, in 2014

                         (a) For each sub-basin

[Figure]

                         (b) For each month
Fig. 11. Comparison of the maximum entropy results between sub-basins and months for each
drought event